# Cancer cell-mitochondria hybrid membrane coated Gboxin loaded nanomedicines for glioblastoma treatment

Yan Zou [1,2], Yajing Sun[1], Yibin Wang[1], Dongya Zhang[1], Huiqing Yang[1], Xin Wang [1], Meng Zheng[1] & Bingyang Shi [1,2] ✉

Glioblastoma (GBM) remains the most lethal malignant tumours. Gboxin, an oxidative phosphorylation inhibitor, specifically restrains GBM growth by inhibiting the activity of $F_0F_1$ ATPase complex V. However, its anti-GBM effect is seriously limited by poor blood circulation, the blood brain barrier (BBB) and non-specific GBM tissue/cell uptake, leading to insufficient Gboxin accumulation at GBM sites, which limits its further clinical application. Here we present a biomimetic nanomedicine (HM-NPs@G) by coating cancer cell-mitochondria hybrid membrane (HM) on the surface of Gboxin-loaded nanoparticles. An additional design element uses a reactive oxygen species responsive polymer to facilitate at-site Gboxin release. The HM camouflaging endows HM-NPs@G with unique features including good biocompatibility, improved pharmacokinetic profile, efficient BBB permeability and homotypic dual tumour cell and mitochondria targeting. The results suggest that HM-NPs@G achieve improved blood circulation (4.90 h *versus* 0.47 h of free Gboxin) and tumour accumulation (7.73% ID/g *versus* 1.06% ID/g shown by free Gboxin). Effective tumour inhibition in orthotopic U87MG GBM and patient derived X01 GBM stem cell xenografts in female mice with extended survival time and negligible side effects are also noted. We believe that the biomimetic Gboxin nanomedicine represents a promising treatment for brain tumours with clinical potential.

Glioblastoma multiforme (GBM), is the most difficult-to-combat cerebral tumours, and presents formidable challenges for effective therapy[1,2]. Currently, the standard of clinical care for GBM is surgical resection, followed by treatment with the GBM first-line drug temozolomide (TMZ) in conjunction with radiotherapy[3]. However, the five-year median survival time of GBM patients is less than 15 months and has not improved significantly in the last decade, highlighting the need for new therapeutic options[4,5]. Gboxin is a well-known inhibitor of oxidative phosphorylation (OXPHOS) mainly by inhibiting $F_0F_1$ ATPase complex V activity in mitochondrial organelles, and thereby inducing the eventual death of GBM tumour cells[6]. Notably, Gboxin specifically suppresses primary GBM cell proliferation with an extremely low half maximal inhibitory concentration ($IC_{50}$) of 150 nM, which is approximately 1000-fold lower than the TMZ (14-250 μM). However, as Gboxin is hydrophobic and unstable, it is quickly eliminated from the body and shows an extremely short elimination half-life of less than 5 min. This factor together with the poor blood brain barrier (BBB) penetration and unspecific internalization, have prevented the successful clinical translation of Gboxin despite its high anti-cancer efficacy[6–8]. Thus, exploration of intelligent delivery systems that transport Gboxin across the BBB and target tumour cells/organelles may help to realize its therapeutic potential in GBM treatment.

[1]Henan-Macquarie University Joint Centre for Biomedical Innovation, Henan Key Laboratory of Brain Targeted Bio-nanomedicine, School of Life Sciences, Henan University, Kaifeng, Henan 475004, China. [2]Centre for Motor Neuron Disease Research, Macquarie Medical School, Faculty of Medicine, Human Health Sciences, Macquarie University, Sydney, NSW 2109, Australia. ✉e-mail: bingyang.shi@mq.edu.au

In recent years, the biomimetic strategy based on natural cell membranes has been utilized to functionalize nanoparticles for targeted delivery of therapeutics agents. Membranes can be derived from various cell types including platelet[9–11], red-blood-cell (RBC)[12–14], leukocyte[15–17], cancer-cell[18,19], stem-cell[20] as well as subcellular-organelles[21]. Membrane camouflaged nanoparticles inherit both the unique physiochemical characteristics of synthetic materials as well as the biological features of the source cells[22–24]. For instance, we and others have demonstrated that RBC membrane cloaking significantly prolongs the plasma circulation time by avoiding induction of immunogenicity[25–27]. Cancer cell membranes have been found to promote homotypic binding, resulting from cell surface interactions mediated by multiple molecules including Thomsen-Friedenreich (TF) antigen and E-cadherin, which elevate the active targeting of nanoparticles[28,29]. Importantly, our very recent report found that GBM cancer cell membrane (CCM) camouflaged nanoparticles possess excellent BBB permeability mediated by down regulation of tight-junction proteins including Zonula occludens-1 (ZO-1), Claudin-5 and Occludin thereby decreasing the tightness of endothelial cells[30]. Additionally, membranes derived from sub-cellular organelles (mitochondria, endoplasmic reticulum etc.) achieve immune escape and can be tailored to contain specific subcellular homotypic targeting proteins. Fusing aim cell and sub-cellular membrane as hybrid to decorate nanoparticles may achieve "two birds, one stone" co-targeting effect where the hybrid biomimetic nanoparticles are specifically taken up by aim cell first and then target the sub-cellular organelles. However, such hybrid membrane driven precise co-targeting strategy have not yet been reported.

In this work, we present a cancer cell-mitochondria hybrid membrane camouflaged reactive oxygen species (ROS)-responsive nanoparticle loaded with Gboxin (HM-NPs@G) to achieve targeting delivery of Gboxin in GBM mitochondria in non-invasive manner. The HM-NPs@G retain characteristic capabilities derived from each individual membrane type. Hence, by design, the outer shell of the HM-NPs@G include multiple "self-marker" proteins embedded in both membranes which should improve the short blood circulation of Gboxin, leading to evasion of immune system clearance. The presence of surface adhesion molecules also should amplify tumour cell and mitochondria co-targeting[31]. We also exploit the fact that mitochondria generate approximately 90% of intracellular ROS and that cancer cells have higher ROS levels than metabolically 'quieter' normal cells to leverage fast, at-site and Gboxin release using a ROS-responsive polymer[32,33]. The accelerated release of Gboxin interrupts the functioning of ATP synthase at the mitochondria inner membrane, which results in disrupted electron transport and energy metabolism ultimately leading to mitochondria-mediated apoptosis in tumour cells[34,35]. We next assess the anticancer efficacy of HM-NPs@G in U87MG and human derived GBM stem cell (GSC, X01) orthotopic xenografts, highlighting the promising potential of our hybrid membrane camouflaging platform for targeted delivery of drugs that cannot be systemically administered.

## Results

### Fabrication of cancer cell-mitochondria hybrid membrane camouflaged nanomedicines with high Gboxin drug loading and ROS responsive drug release

The fabrication of HM-NPs@G consists of two steps (Fig. 1a): first, the outer shell of cancer cell-mitochondria hybrid membrane (HM) was prepared using a 1:1 protein weight ratio of MM (mitochondria membrane) to CM (cancer membrane) as optimized and further characterized by förster resonance energy transfer (FRET)[36]. In terms of the ratio of MM between inner and outer membrane, it may be the same as the natural mitochondria because we isolated the total mitochondrial membranes containing both inner and outer membranes. The core-shell structure of the developed HM-NPs@G was confirmed with transmission electron microscopy (TEM) (Fig. 1b), also indicating the hybrid membrane is a single-membrane lipid bilayer which is agree

with the reported results[37,38]. The successful fusion of CM and MM was demonstrated by confocal microscopy as co-localization of specific CM (red) and MM (green) fluorescent signals were observed (Fig. 1c). Afterwards, we have further characterized proteins on the CM and MM by the western blots[18,39,40]. As shown in Fig. 1d, the key proteins (Atlastin-1, EHD2 and Mito-fusion) related to mitochondria targeting and penetration were observed on MM and HM-NPs. In addition, the proteins (EpCAM and Integrin αv) which play vital roles in cancer homologous targeting were observed on U87MG cancer cell membrane (CM). Furthermore, glioblastoma stem cell (GSCs, X01) membrane CM (X01) had CD44, one of stem markers, as well as EpCAM, both of which were helpful to target homotypic cells. Surprisingly, CD44 and Integrin αv also expressed on MM, which endow the MM-NPs targeting capability to GBM cells to some extent (Fig. 1d). Moreover, Atlastin-1, which is closely related to the bio-membrane fusion, was expressed on CM, MM and HM-NPs, facilitating the permeability to tumour cell and mitochondria. Second, the inner core was fabricated from Gboxin loaded ROS-responsive polymeric nanoparticles based on poly (ethylene glycol)-poly (4-(4, 4, 5, 5-Tetramethyltetramethyl-1, 3, 2-dioxaborolan-2-yl) benzyl acrylate) (PEG-PHB) (Supplementary Figs. 1–3) which was subsequently decorated with the HM. It should be noted that the PEG-PHB polymer could be degraded into PEG, pinacol borate and p-hydroxy-methylphenol and further eliminated from the body. Furthermore, gel electrophoresis analysis of membrane protein markers indicated good retention of characteristic MM and CM proteins in the protein profile of HM and HM-NPs (Supplementary Fig. 4), confirming that it is indeed possible to fuse two different types of cell membranes and engraft onto the same nanoparticle[41]. Western blotting analysis (Supplementary Fig. 5) also showed that Bcl-2 was observed on MM, HM and HM-NPs, further confirming MM characteristic proteins coated on the surface of nanoparticles. Considering Gboxin is difficult to envelop in polymer due to its special physical and chemical properties, we assessed four different polymer types for optimal interaction. PEG-PHB polymer showed the highest interaction energy with Gboxin (−25.1 kcal/mol) among the four designed polymers (Fig. 1e). Hence, PEG-PHB polymer was chosen for loading and delivering Gboxin in the further study. The corresponding HM-NPs@G showed a high Gboxin loading content of 15.0% with loading efficiency of 70.4% (Supplementary Table 1). As shown by dynamic light scattering (DLS), bare nanoparticles had an original size of 63.9 nm, which increased in size by 26-29 nm after coating with single cancer cell membrane (CM-NPs@G, 93.3 nm), mitochondrial membrane (MM-NPs@G, 88.2 nm) or fused HM (89.1 nm), respectively (Supplementary Table 2, Fig. 1f). In addition, surface charge changed from +4.1 mV to −25.6 mV after the NPs were coated with membranes (Fig. 1g), indicating successful shielding of nanoparticles by the negative outer membranes.

We next evaluated the in vitro release profile of HM-NPs@G in a media containing $H_2O_2$ that mimics the intracellular ROS environment to determine the effect of ROS on release kinetics. Cumulative Gboxin release was 86.6% and 50.2% from HM-NPs@G in the presence of 1 mM or 0.1 mM $H_2O_2$ after 24 h incubation, respectively (Fig. 1h), which was in marked contrast to the 15.4% Gboxin release achieved in the absence of $H_2O_2$ mimicking non-oxidative physiological conditions. Moreover, the size and polydispersity index (PDI) of HM-NPs@G showed noticeable increases when exposed to $H_2O_2$ in comparison to nanoparticles (NPs) under normal physiological conditions (Fig. 1i). Collectively, these results suggest that HM-NPs@G are ROS-responsive, leading to controlled Gboxin release under conditions that mimic the high ROS tumour environment.

### Enhanced cellular internalization, mitochondria targeting and anti-tumour efficacy

Cellular internalization and intracellular release of NP cargoes play a vital role in enhancing drug bioactivity[42,43]. Cellular uptake of

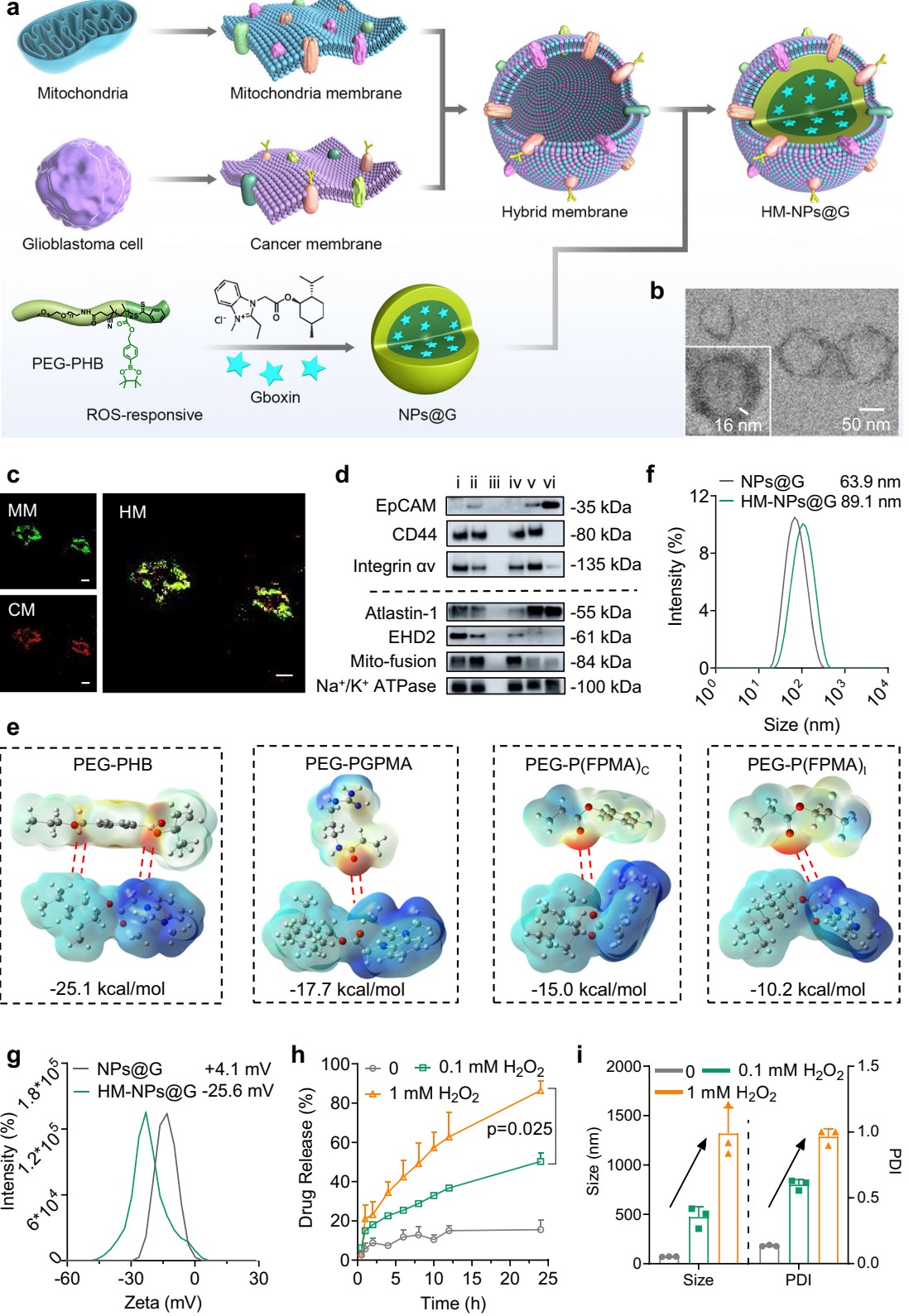

HM-NPs was evaluated by loading NPs with Cyanine 5 (Cy5) or fluorescein isothiocyanate (FITC) with subsequent detection by flow cytometry as well as confocal laser scanning microscopy (CLSM) in homologous U87MG GBM cells. HM-NPs showed a 2-fold higher Cy5 intensity than that produced by naked nanoparticles (Fig. 2a). CLSM images showed that HM-NPs produced obviously enhanced green fluorescence in the cytoplasm of U87MG cells

compared with NPs without membrane modification (Fig. 2b, left). Collectively, these results demonstrated that HM-NPs had better cellular internalization capability. To further investigate homologous mitochondrial targeting, we assessed whether co-localization of NPs with U87MG tumour cell mitochondria occurred using MitoTracker red probe. Treatment of U87MG cells with HM-NPs and MM-NPs produced enhanced yellow fluorescence

**Fig. 1 | Fabrication and characterization of HM-NPs@G, Gboxin loading and controlled release. a** Fabrication of cancer cell-mitochondria hybrid membrane camouflaged Gboxin encapsulated ROS-responsive polymeric nanoparticles (HM-NPs@G). **b** Transmission electron microscope (TEM) images of HM-NPs@G. Scale bar = 50 nm. The TEM images were representative data from three independent experiments. **c** CLSM images of fabricated hybrid membrane (HM) vesicles. Mitochondrial membranes (MM) were labeled with DiO (green) and cancer cell membranes (CM) were labeled with DiD (red). The merged image showed yellow fluorescence with similar morphology to MM and CM confirming the successful fabrication of HM vesicles. Scale bar = 20 μm. The CLSM images were representative data from three independent experiments. **d** Western blotting analysis cancer membrane and mitochondria membrane special targeting related proteins. i: HM-NPs (X01), ii: HM-NPs (U87MG), iii: NPs, iv: MM, v: CM (X01), vi: CM (U87MG). The immunoblots were representative data from three independent experiments. **e** Molecular electrostatic potential mapping (MEP, C atoms are grey, N atoms are blue, O atoms are red, S atoms are pink, F atoms are light-blue and H atoms are white) of the four polymers PEG-PHB, PEG-PGPMA, PEG-P(FPMA)$_C$ and PEG-P(FPMA)$_I$) with Gboxin where the color changes from red to blue (red represents negative electrostatic potential and blue represents positive electrostatic potential) showing Gboxin loading via electrostatic interaction. **f** Size distribution of NPs@G and HM camouflaged HM-NPs@G. The size analysis was representative data from three independent experiments. **g** Zeta potential of NPs@G and HM-NPs@G determined by dynamic light scattering. The Zeta analysis was representative data from three independent experiments. **h** Cumulative Gboxin release as well as (**i**) the change in size and PDI of HM-NPs@G in phosphate buffer (PB) containing $H_2O_2$ (0.1 mM and 1 mM) at 37 °C, PB without $H_2O_2$ was used as a control. The drug release and change in size and PDI analyses were representative data from three independent experiments. Data are presented as mean ± SD. Source data are provided as a Source Data file.

(Fig. 2b, right) resulting from the overlap of red (mitochondria) and green (NPs), whereas U87MG cells treated with only cancer membrane coated nanoparticles (CM-NPs) did not produce marked yellow fluorescence. These results indicate that HM-NPs and MM-NPs have preferable mitochondria targeting capability. Co-localization line scanning profile calculated using CLSM software also supports this conclusion (Fig. 2c). Furthermore, the targeting ability of HM was further assessed by treating cells with HM encapsulating upconversion nanoparticles (HM-UCNPs) and being observed with Bio-TEM. The results showed that notable UCNPs were delivered into U87MG cells by HM-UCNPs and CM-UCNPs with active targeting of CM, while much fewer UCNPs were observed in MM-UCNPs and negligible UCNPs were observed in the naked UCNPs treating cells (Supplementary Fig. 6), indicating the homologous targeting capability of CM. Meanwhile, single CM modified UCNPs showed the limited capability to target and accumulate in the mitochondria, evidenced by abundant UCNPs in the cytoplasm rather than the mitochondria. Interestingly, the majority of HM-UCNPs were located in the mitochondria, suggesting the dual-targeting of HM to both tumour cells and mitochondria organelles. Collectively, these results indicate that the hybrid membrane coating strategy generates biomimetic NPs with superior homologous tumour cell and mitochondria targeting. Moreover, to reveal the subcellular targeting mechanism of HM-NPs@G, we selected a mitofusin inhibitor MFI8 to evaluate the Gboxin content in mitochondria as the mitofusin protein has been reported to play a key role in fusion of mitochondria membrane and mainly expresses on the MM. The results show that the accumulation of HM-NPs@G in mitochondria isolated from U87MG cells pre-treated with MFI8 remarkably reduced compared to the group without pre-treatment, suggesting that mitofusin plays a key role in targeting and penetrating the mitochondria (Supplementary Fig. 7). However, the mitochondria-targeting mechanism is very complicated. We briefly demonstrate that mitofusin is involved in the mitochondria targeting of HM-NPs, and the systematic targeting mechanism deserves further investigation.

We next assessed the ability of NPs to inhibit proliferation of U87MG and X01 GBM stem cells (GSCs) using the CCK-8 assay. Treatment with HM-NPs@G resulted in the most potent inhibition of cell proliferation in both U87MG and X01 cells, compared to treatment with free Gboxin, NPs@G, CM-NPs@G or MM-NPs@G (Fig. 2d, Supplementary Fig. 8). Additionally, biomimetic HM-NPs@G NPs showed that they could effectively boost the cytotoxicity of Gboxin to GBM cells but also reduce toxicity to normal cells, as evidenced by much higher IC$_{50}$ values (5-10 folds) in normal cells but significantly lower IC$_{50}$ values in GBM cells (Fig. 2e, Supplementary Fig. 9). To further validate the cytotoxicity of HM-NPs@G, we used the Annexin V-FITC apoptosis assay and results showed that HM-NPs@G induced 43.17% and 55.15% cell apoptosis in U87MG and X01 cells, respectively (Supplementary Figs. 10, 11), which was consistent with the CCK-8 results and showed the ability of HM-NPs@G to promote cell apoptosis. Collectively, HM-NPs@G NPs, with both tumour and mitochondria targeting capability, represents a distinct advantage in delivering Gboxin to GBM cells and targeting GBM mitochondria that subsequently interferes with cell proliferation.

Gboxin, an inhibitor of oxidative phosphorylation, can inactivate ATP synthase in the mitochondrial intima and is strongly associated with ATP synthesis and electron transport chains[6,44]. Inactivation of ATP synthase inhibits the synthesis of ATP and promotes mitochondrial depolarization[44]. Hence, we assessed ATP levels and mitochondrial membrane potential using the ATP and JC-1 detection kit. HM-NPs@G resulted in a sharp reduction of ATP levels in both U87MG and X01 cells (Fig. 2f, Supplementary Fig. 12). After that, in normal mitochondria, JC-1, a lipophilic cationic dye, aggregates to emit red fluorescence but when mitochondrial membrane potential (Δψm) is reduced, JC-1 becomes dispersed and adopts a monomeric form which produces a green fluorescence[45,46]. Accordingly, a strong green fluorescence was produced in GBM cells by treatment with HM-NPs@G indicating a decrease in Δψm (Fig. 2g, h). In contrast, strong red and weak green fluorescence were observed after treatment with free Gboxin, NPs@G, CM-NPs@G or MM-NPs@G demonstrating relatively healthy mitochondria (Supplementary Fig. 13). In addition, the oxygen consumption rate (OCR) was conducted in U87MG and X01 cells. Both U87MG cells and X01 cells were subjected to different formulations include free Gboxin, NPs@G, CM-NPs@G, MM-NPs@G, and HM-NPs@G. The results showed that HM-NPs@G led to the OCR of 18.87% and 10.44% for U87MG and X01 cells, which were the lowest as compared with other groups (Supplementary Fig. 14), suggesting that superior anti-tumour effects of HM-NPs@G benefit from the consumption of oxygen. Intriguingly, much lower OCR was detected in X01 than U87MG cells, indicating the sensitivity of GSCs cells to Gboxin and in accordance with the lower IC$_{50}$ concentration. Mitochondrial depolarization can result in the translocation of Cyto C from the intermembrane space to the cytosol, a process which is considered as a key inducer of cell apoptosis. Bio-TEM was then utilized to detect mitochondrial structure. Treatment of GBM cells with HM-NPs@G resulted in conspicuous damage to mitochondrial structures (Fig. 2i). To further identify the structure of mitochondria, we used the microglia cell as a control to demonstrate the typical mitochondria morphology that is significantly different from damaged mitochondria in U87MG cells induced by the HM-NPs@G (Supplementary Fig. 15), indicating the therapeutic effect of HM-NPs@G. A known consequence of the translocation of Cyto C to the cytosol is the initiation of a cascade of caspase reactions. Accordingly, we assessed the expression of Cyto C, caspase 3/9 (C-3/9) and cleaved caspase 3/9 (CC-3/9) proteins in U87MG cells by western blotting. These results showed that Cyto C was up-regulated by HM-NPs@G treatment compared to monomembrane coated NPs (CM-NPs@G and MM-NPs@G), bare NPs@G and was significantly up-regulated relative to free Gboxin treatment

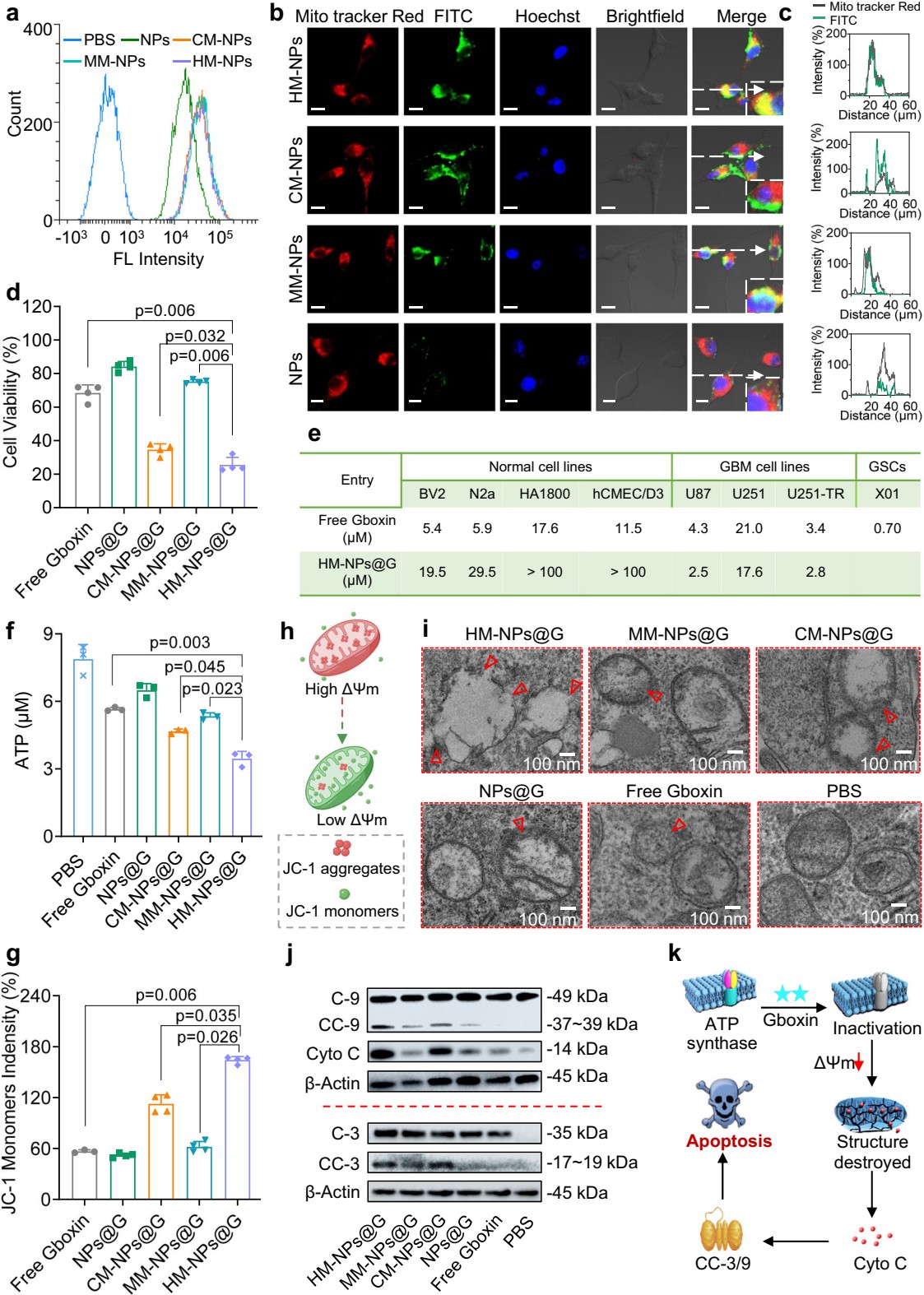

(Fig. 2j, k). Considerable enhancement in the expression of C-3/9 and cleaved CC-3/9 were also observed after HM-NPs@G treatment, suggesting that intrinsic apoptosis was activated by Gboxin released from HM-NPs@G. Taken together, it can be concluded that cell apoptosis was regulated by the activation of intrinsic mitochondrial apoptotic caspase pathway, which increased the inactivation of ATP synthase and elevated GBM tumouricidal effects.

## On-target effects of HM-NPs@G assay in vitro

The on-target effects of HM-NPs@G were evaluated by isolating and quantifying the Gboxin in mitochondria determined by High Performance Liquid Chromatography (HPLC). The results showed that about 23.4 µg/mL Gboxin was accumulated in the mitochondria of HM-NPs@G treated cells, which was significantly higher than that of single membrane coated nanomedicines CM-NPs@G (16.6 µg/mL) and

**Fig. 2 | Enhancement of specific GBM cell inhibition by HM-NPs@G. a** Flow cytometry analysis of U87MG cell targeting following 6 h incubation with Cy5 loaded HM-NPs, CM-NPs, MM-NPs or NPs (Cy5 concentration was 10 μg/mL). The flow cytometry analyses were representative data from three independent experiments. CLSM images (**b**) and (**c**) co-localization analysis of U87MG cells incubated with HM-NPs, CM-NPs, MM-NPs or NPs for 6 h. (Mitochondria stained by Mito tracker Red (Red), Nuclei stained by Hoechst (blue), FITC-labeled NPs (green)). The concentration of FITC was 10 μg/mL. Scale bar = 10 μm. The CLSM images and co-localization analysis were representative data from three independent experiments. **d** U87MG cell viability after 72 h incubation with free Gboxin, NPs@G, CM-NPs@G, MM-NPs@G, or HM-NPs@G ($n = 4$ biologically independent samples). Data are presented as mean ± SD (one-way ANOVA and Tukey's multiple comparison test). **e** $IC_{50}$ values in normal cells (N2a, BV2, HA1800 and hCMEC/D3 cells) and GBM cells (U87MG, U251, U251-TR and X01) after 72 h incubation with HM-NPs@G or free Gboxin ($n = 4$ or 5 biologically independent samples, exact n seen in Supplementary Fig. 9). **f** ATP concentrations in U87MG cells treated with free Gboxin, NPs@G, CM-NPs@G, MM-NPs@G, or HM-NPs@G for 72 h ($n = 3$ biologically independent

samples). Data are presented as mean ± SD (one-way ANOVA and Tukey's multiple comparison test). **g** Quantitative analysis of JC-1 monomer fluorescence intensity in U87MG cells treated with free Gboxin, NPs@G, CM-NPs@G, MM-NPs@G, or HM-NPs@G for 72 h ($n = 4$ biologically independent samples). Data are presented as mean ± SD (one-way ANOVA and Tukey's multiple comparison test). **h** Schematic illustration of JC-1 structure changes resulting from changes in mitochondrial membrane potential changes. **i** TEM images of mitochondria in U87MG cells treated with PBS, free Gboxin, CM-NPs@G, MM-NPs@G or HM-NPs@G. The red arrows indicated the damaged mitochondria structure after treatment of nanomedicines. The TEM images were representative data from three independent experiments. **j** Western blotting analysis of cytochrome c (Cyto C) and apoptosis-related proteins treated with free Gboxin, NPs@G, CM-NPs@G, MM-NPs@G, or HM-NPs@G for 72 h. The immunoblots were representative data from three independent experiments. **k** Schematic illustration of the mechanism of HM-NPs@G mediated GBM cell apoptosis. For Fig. 2d and Fig. 2f–j, the concentration of Gboxin was 800 nM. Source data are provided as a Source Data file.

MM-NPs@G (12.1 μg/mL), while Gboxin could not be detected for the free Gboxin and NPs@G treatments (Supplementary Fig. 16). These all indicate the good homotypic targeting of hybrid membrane coated nanomedicines and the on-target effects directly benefit to the tumour cell growth inhibition.

### HM-NPs@G improves Gboxin in vivo pharmacokinetics and achieves GBM tumour tissue and mitochondria dual-targeting

Prior to evaluating the in vivo therapeutic effects of Gboxin-loaded NPs, we first investigated the pharmacokinetics of HM-NPs@G in healthy mice. Gboxin blood levels were monitored to estimate plasma clearance kinetics after a single intravenous (i.v.) injection via tail vein. Free Gboxin showed an evidently short blood circulation (t$_{1/2}$, β) of 0.47 h, indicating rapid elimination and could not be detected after 4 h (Fig. 3a). Bare NPs@G showed a longer circulation with a half-life of 1.5 h, which may reflect prevention of rapid clearance from the blood due to the pegylated neutral shell used in formulating the NPs. MM-NPs@G showed an enhanced blood circulation time of 2.7 h. Notably, HM-NPs@G and CM-NPs@G showed similarly improved half-lives at approximately 4.9 h, suggesting that membrane camouflaging helps NPs escape from immune recognition and subsequent clearance in blood, presumably due to the presence of 'self' recognizable proteins on the membrane as previously reported[47,48]. The improved pharmacokinetics of HM-NPs@G due to homologous camouflaging was expected to facilitate accumulation in tumour and mitochondria. To confirm whether cancer membrane camouflaging promoted specific NPs accumulation in GBM, near-infrared dye Cy5 loaded NPs were systemically injected into luciferase expressing U87MG (U87MG-Luc) tumour-bearing nude mice and monitored in real-time with Cy5 fluorescence. An obviously stronger red fluorescence was observed in the brain of mice treated with HM-NPs and CM-NPs 6 h post-injection which remained detectable up to 24 h (Fig. 3b), indicating that both HM-NPs and CM-NPs were able to traverse the BBB to accumulate in GBM tissue[28]. In contrast, only a weak fluorescence was observed in the brain after treatment with MM-NPs or naked NPs. Treatment with free Cy5 resulted in rapid loss of fluorescence which could not be detected at 8 h post injection, indicating that protection with nanocarriers and membrane camouflage facilitate GBM accumulation due to prolonged blood retention.

We also confirmed real-time Cy5 imaging results by evaluating the accumulation of HM-NPs in the main organs (including heart, liver, spleen, lung, kidney, tumour bearing brain) of U87MG-Luc orthotopic xenografts ex-vivo. Compared with free Cy5, MM-NPs showed a relatively strong fluorescence signal located in the tumour sites, which reflected better retention of nanocarriers. Importantly, HM-NPs and CM-NPs exhibited an obviously stronger fluorescence in tumour sites, suggesting that the camouflage of cancer membrane enhanced

accumulation (Fig. 3c, Supplementary Fig. 17). Importantly, the fluorescence of HM-NPs in the brain showed marked co-localization with tumour luminescence (Fig. 3c). In terms of Gboxin biodistribution, as measured by HPLC, the brain tumour accumulation of Gboxin after treatment with HM-NPs@G and CM-NPs@G was 7.73% and 7.43% of injected dose per gram of tissue (%ID g$^{-1}$), respectively, which was approximately 1.72- and 6.90-fold higher than that achieved by MM-NPs@G or free Gboxin (Fig. 3d). Given the high levels of HM-NPs detected in the kidney, we considered that HM-NPs could be metabolized and excreted out of the body by renal corpuscles, which further, at least partly confirmed by the results of renal distribution of HM-NPs (Supplementary Fig. 18).

We further investigated NPs co-localization with tumour mitochondria. Mice bearing U87MG-Luc orthotopic xenografts were treated with Cy5 loaded HM-NPs with brains collected at 6 h post-injection. Tumours excised from HM-NPs treated mice exhibited stronger yellow fluorescence (resulting from an overlay of red Cy5-loaded nanoparticles and green Anti-Hsp60 Rabbit pAb dyed mitochondria) in GBM tissue, indicating that the co-localization efficacy of HM-NPs and mitochondria was significantly higher than that of other treatments (Fig. 3c, bottom), reflecting the homologous targeting of the mitochondrial membrane. Taken together, these results clearly demonstrated that HM-NPs not only have longer blood circulation time but also active homotypic tumour and mitochondria dual-targeting capability.

### Mechanistic aspects of BBB and GBM tissue penetration by HM-NPs

Firstly, tumour penetration by HM-NPs was evaluated in U87MG multicellular spheroids (Fig. 4a). At a scanning depth of 60 μm, both naked and MM camouflaged nanoparticles failed to produce significant fluorescence which was localized to the periphery of multicellular spheroids (Fig. 4b). In contrast, fluorescence produced by HM-NPs treatment was clearly observable inside the multicellular spheroids even at a scanning depth of 80 μm and was confirmed quantitatively (Fig. 4c). These results again highlighted the penetration capability of HM-NPs.

There are multiple interaction molecules on the surface of cancer membrane including integrin, Mac-1 and other special proteins, which facilitate the membrane coated nanoparticles to traverse the BBB by modulating the tight junctions[31,37,49]. BBB disruption is a well-recognized mechanism of elevated BBB penetration mediated by brain metastatic cells[50,51]. Given that the tight junctions between endothelial cells are highly associated with BBB disruption[31,52], We determined if GBM cell membrane decorated NPs can traverse the BBB by modulating the tight junctions between endothelial cells using an in vitro BBB model consisting of a top layer of human cerebral

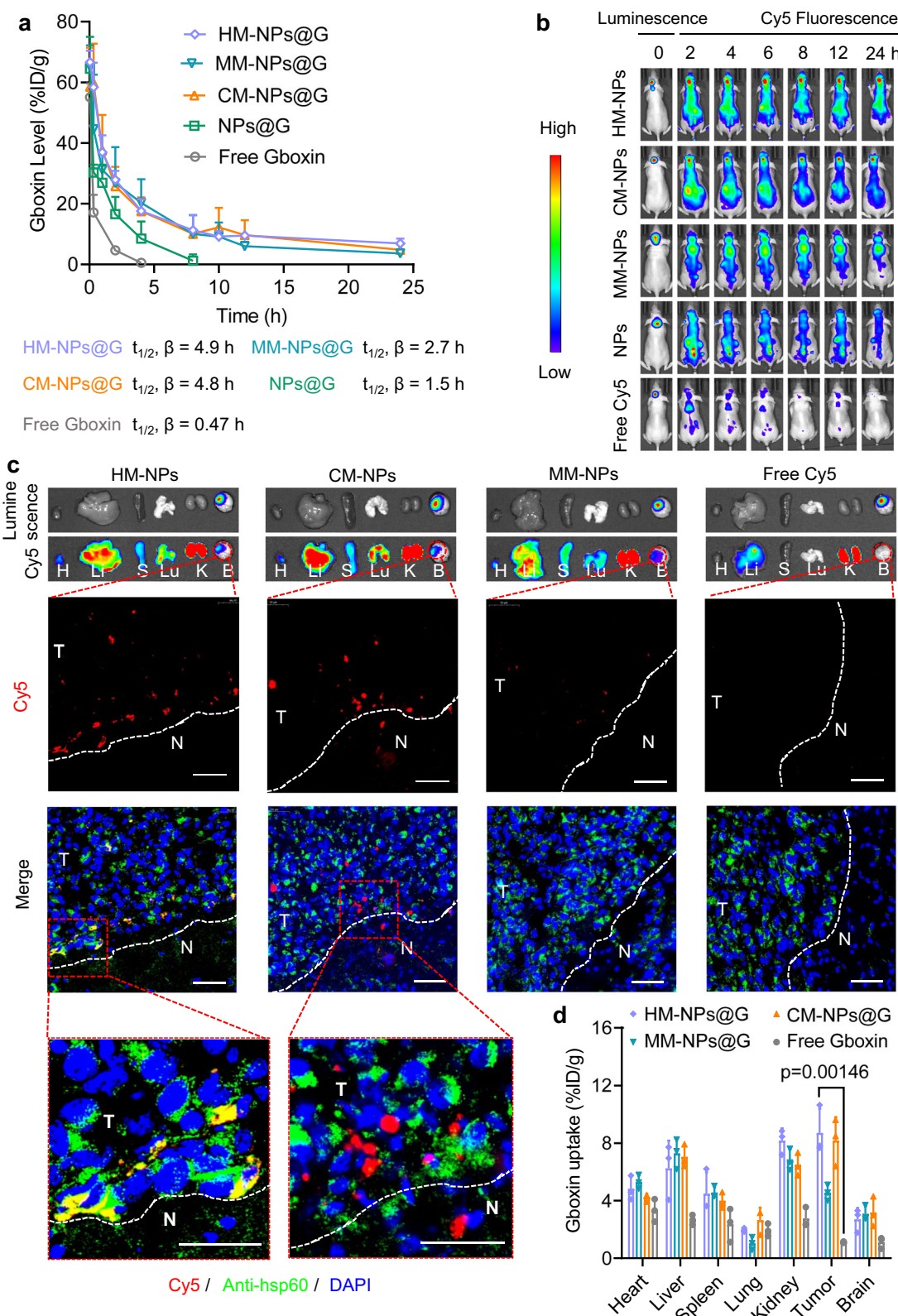

Cy5 / Anti-hsp60 / DAPI

microvascular endothelial cells (hCMEC/D3) as well as astrocytes (HA1800 cells) and lower layer of U87MG cells. To assess the BBB model integrity, trans-endothelial electrical resistance (TEER) was continuously monitored after nanoparticles were added to the upper compartment. In this model, HM-NPs enhanced BBB traversal was observed (Fig. 4d, e) with evidence that the TEER of the hCMEC/D3 and HA1800 bilayer decreased after HM-NPs or CM-NPs treatment (Fig. 4f),

whereas treatment with MM-NPs, naked NPs or PBS showed little, or no, reductions in TEER values. Significantly, flow cytometry showed increased Cy5 fluorescence intensity in U87MG cells harvested from the lower compartment after treatment with HM-NPs or CM-NPs (Supplementary Fig. 19a), suggesting that GBM cell membrane endowed NPs with BBB traversal capability in homologous U87MG cells. Subsequently, we pretreated hCMEC/D3 cells with a combination

**Fig. 3 | In vivo pharmacokinetics, BBB penetration, GBM and mitochondrial targeting by HM-NPs. a** Pharmacokinetic profiles of HM-NPs@G, CM-NPs@G, MM-NPs@G or NPs@G in healthy BALB/c mice with free Gboxin as control (5 mg Gboxin equiv. kg$^{-1}$, $n = 3$ mice in each group). Data are presented as mean ± SD. **b** In vivo fluorescence images of orthotopic U87MG-Luc human GBM tumour bearing nude mice following a single tail vein injection of HM-NPs@Cy5 (2 mg Cy5 equiv. kg$^{-1}$). The images were representative data from three mice. **c** Bioluminescence and Cy5 fluorescence images of major organs and tumour tissue taken from U87MG-Luc bearing mice 6 h post injection of HM-NPs@Cy5 (1 mg Cy5 equiv. kg$^{-1}$). H heart; Li liver; Lu lung; S spleen; K kidney; B brain. Enlarged images were captured by CLSM to demonstrate tumour tissue targeting, tumour penetration as well as co-localization with mitochondria in GBM cells after treatment with HM-NPs@Cy5 and controls (CM-NPs@Cy5, MM-NPs@Cy5 or free Cy5). Nuclei were stained with DAPI (blue) and mitochondria with Anti-hsp60 (green); Cy5 fluorescence is red. Dotted lines indicate orthotopic GBM boundaries (N normal brain tissue, T tumour tissue). Scale bars = 50 μm. The distribution analyses were representative data from three mice. **d** Quantification of Gboxin accumulation in major organs and orthotopic GBM tissue excised from mice 6 h post tail vein injection of with HM-NPs@G (5 mg Gboxin equiv. kg$^{-1}$). Gboxin levels were determined by HPLC and expressed as injected dose per gram of tissue (%ID g$^{-1}$) ($n = 3$ mice in each group). Data are presented as mean ± SD (one-way ANOVA and Tukey multiple comparisons tests). Source data are provided as a Source Data file.

of 8-(4-chlorophenylthio)-adenosine 3′,5′-cyclic monophosphate (8-CPT-cAMP) and 4-(3-butoxy-4-methoxybenzyl)-2-imidazolidinone (RO-20-1724), which enhance the expression of junction proteins[53]. Interestingly, no alteration in TEER values (Fig. 4g) or BBB penetration efficiency (Supplementary Fig. 19b) was observed after treatment with HM-NPs, MM-NPs, CM-NPs, NPs or PBS, indicating that BBB integrity was retained and further highlighting the role played by junction proteins in the mechanism of BBB penetration.

As it has been reported that ZO-1 and claudin-5 in endothelial cells play critical roles in regulating tight junctions[54,55], we next assessed the expression of ZO-1 and claudin-5 in hCMEC/D3 monolayers (Fig. 4h). Expression levels of ZO-1 and claudin-5 in hCMEC/D3 monolayers after treatment with HM-NPs and CM-NPs for 48 h were markedly lower than that mediated by MM-NPs, naked NPs or PBS (Fig. 4i), indicating that GBM cell membrane may reduce the tightness of tight junction. Moreover, after 72 h treatment, expression of ZO-1 and claudin-5 did not show significant difference between treatments (Fig. 4j), indicating the recovery of tight junctions. To provide evidence of the modulation of tight junctions by GBM membrane decorated NPs in vivo, immunofluorescence staining of ZO-1 in GBM brain tissue was performed. These results showed that ZO-1 expression was significantly down-regulated in the tumour zone after 48 h treatment with HM-NPs and CM-NPs (Fig. 4k). In agreement with in-vitro western blotting results, there was no obvious difference in ZO-1 expression after 72 h treatment with HM-NPs, MM-NPs and CM-NPs (Fig. 4l), suggesting the quick regeneration of BBB tight junctions. Collectively, these results further confirmed that HM-NPs effectively crossed the BBB by modulating tight junctions between endothelial cells.

### In vivo anticancer effects of NPs treatment in GBM xenografts

In order to evaluate the antitumour efficacy of HM-NPs@G, mice with established orthotopic U87MG-Luc tumour were treated with 3 mg Gboxin equiv.kg$^{-1}$ via i.v. injection every three days for five doses in total (Supplementary Fig. 20a). Body bioluminescence as well as body weight was monitored during the treatment period. Mice receiving HM-NPs@G treatment exhibited minimal increase in GBM luminescence signal (Supplementary Fig. 20b). Treatment with MM-NPs@G or CM-NPs@G also presented markedly increased tumour luminescence, whereas free Gboxin or PBS treatment showed exponential signal increase. Histological examination by hematoxylin and eosin (H&E) staining of the whole brain further confirmed that HM-NPs@G treatment resulted in the smallest tumour size (Supplementary Fig. 20c). Body bioluminescence measurements were also confirmed quantitatively (Supplementary Fig. 20d). Furthermore, the average body weight of mice following HM-NPs@G treatment was comparatively stable, indicating negligible systemic toxicity (Supplementary Fig. 20e). However, marked decreases in body weight were observed after treatment with free Gboxin or PBS, indicating increased brain damage as GBM tumour grew. In terms of survival, HM-NPs@G resulted in the longest median survival time (67.5 days) which was significantly longer than that of PBS (36.5 days), free Gboxin (35 days), CM-NPs@G (43.5 days) or MM-NPs@G (40 days) (Supplementary Fig. 20f). In addition, the anticancer mechanism was examined by western blotting (Supplementary Fig. 20g) in excised tumour tissues. The expressions of Cyto C, C-3/9 and CC-3/9 were significantly up-regulated in GBM tumour tissue from mice receiving HM-NPs@G, demonstrating that tumour inhibition resulted from the mitochondria-dependent apoptosis pathway. TUNEL immunofluorescence analysis confirmed the increased tumour cell apoptosis induced by HM@NPs (Supplementary Fig. 20h). Immuno-histochemical staining for CC-3 and Ki-67 were in line with protein expression results. Expression of CC-3 was significantly higher after HM-NPs@G treatment compared to that with single membrane MM-NPs@G or CM-NPs@G (Supplementary Fig. 21). Conversely, expression of the proliferation marker Ki-67 was the lowest in mice receiving of HM-NPs@G. These results collectively indicated that hybrid membrane coated nanomedicines had superior anti-tumour effects in orthotopic GBM mice models. Interestingly, H&E staining indicated that kidney injury was induced by free Gboxin treatment, while no toxic side effects were evident in the major organs of mice receiving multiple-doses of Gboxin-containing nanoparticles (Supplementary Fig. 20c, 22).

### In vivo anticancer effects of NPs treatment in GBM stem cell xenografts

Although HM-NPs@G nanomedicines have shown good therapeutic effects on U87MG orthotopic mice models, it is admitted that U87MG model has a couple of limitations including the unclear originals. Therefore, the patient derived xenograft (PDX) glioblastoma stem cells (GSCs) models are adopted as they are more closely resemble clinical GBM characteristics than standard GBM cell derived xenograft (CDX) models. Next, we investigated the anti-tumour effect of HM-NPs@G in X01 patient-derived GSCs xenograft models. To test whether HM-NPs@G effectively inhibit growth of GSCs in vivo, we developed luciferase-expressing X01 cells (X01-Luc) to establish orthotopic GSCs mouse models (Fig. 5a). After the same treatment schedule, consistent outcomes to the previously used orthotopic U87MG were found. Specifically, HM-NPs@G treatment significantly inhibited tumour growth as shown by minimal increase in GSCs luminescence signal (Fig. 5b). Stable body weight was also observed in mice receiving HM-NPs@G treatment, further demonstrating potent anti-tumour effects and negligible side effects which were in marked contrast to the large loss of body weight resulting from PBS or free Gboxin treatments (Fig. 5c). As a result of tumour growth inhibition, the survival rate of HM-NPs@G treated mice was extended by 80% at 36 days (Fig. 5d). Quantified luminescence further verified the enhanced therapeutic effects of HM-NPs@G (Fig. 5e) as confirmed by H&E staining of excised brain from X01-Luc tumour bearing mice which showed that mice receiving HM-NPs@G had the smallest tumour volume (Fig. 5f, Supplementary Fig. 23). In addition, the western blots confirmed that the expression of Cyto C, C-3/9 and CC-3/9 were markedly up-regulated in tumour tissues excised from mice treated with HM-NPs@G (Fig. 5g). TUNEL analysis further demonstrated substantial apoptosis and decreased proliferation in GSC tumour after HM-NPs@G treatment (Fig. 5h) and H&E staining further highlighted kidney damage caused

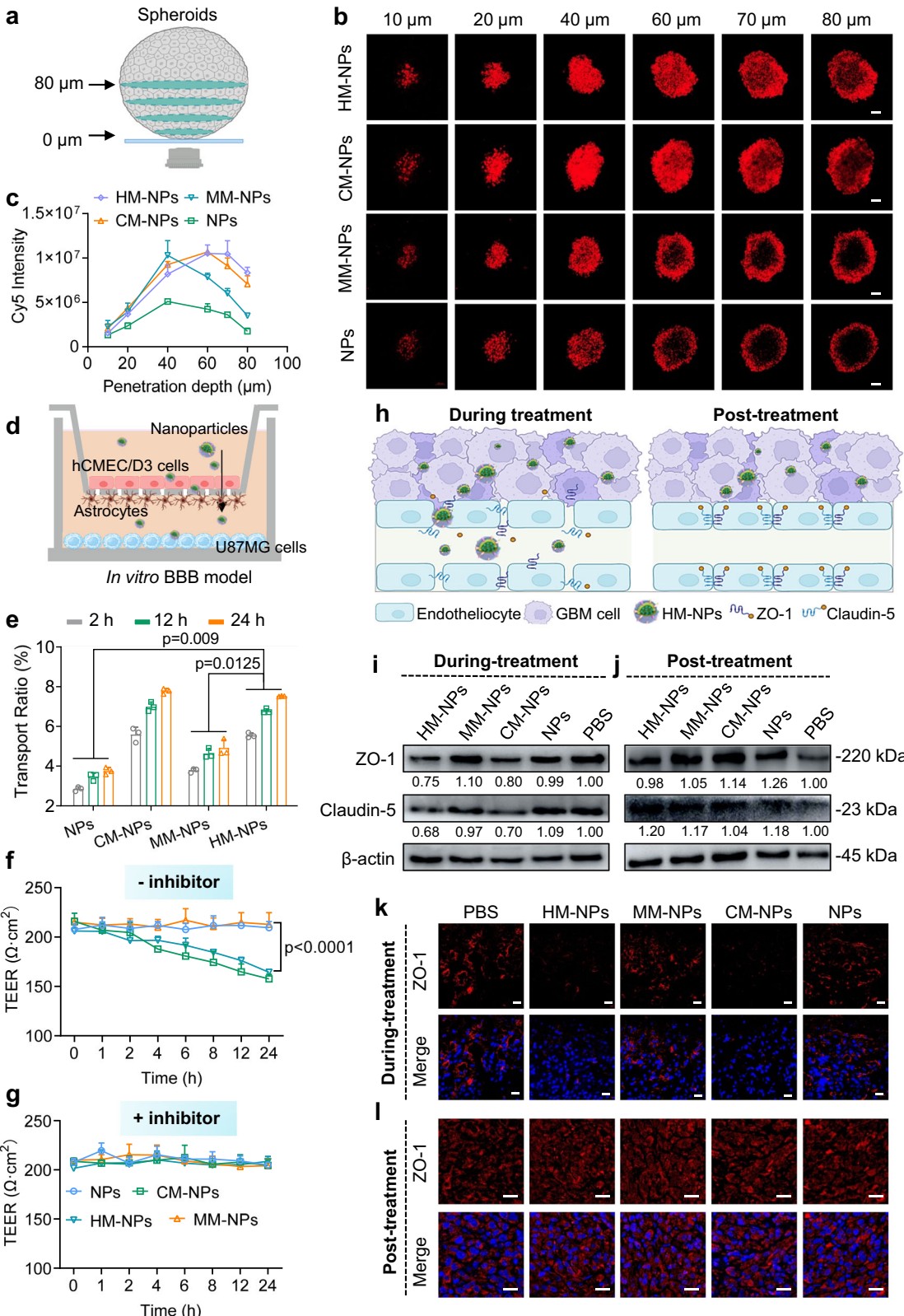

by free Gboxin (Fig. 5i). Collectively, these results confirmed that hybrid membrane decorated nanomedicines had promising potential in suppressing tumour growth in patient-derived GSCs xenograft models.

In order to compare the anti-GBM effect of HM-NPs@G and GBM first-line drug TMZ, we have performed the treatment of HM-NPs@G towards orthotopic GSC models by using the TMZ as a control via oral administration to mimic the clinical treatment (Fig. 6a). The results showed that HM-NPs@G had a significantly enhanced anti-tumour effect compared with free TMZ. Interestingly, though the tumours could be restrained to some extent after being treated with free TMZ, the tumours erupted promptly when the injection was terminated. In sharp contrast, HM-NPs@G group exhibited continuous tumour inhibition in a longer period (Fig. 6b, c). The body weight of mice had a

**Fig. 4 | GBM tissue penetration and mechanism of BBB penetration by HM-NPs.**
**a** Schematic illustration of the in vitro 3D spherical tumour model. **b** Penetration, and, (**c**), quantitation of Cy5 loaded HM-NPs distribution in U87MG multicellular spheroids after 6 h incubation (Cy5 concentration was 10 μg/mL). Scale bars = 100 μm (*n* = 3 biologically independent samples). Data are presented as mean ± SD. **d** Illustration of the in vitro BBB model. **e** Transport ratios of bare NPs, CM-NPs, MM-NPs or HM-NPs in hCMEC/D3 monolayer of in vitro BBB model (Cy5 concentration: 10 μg/mL). (*n* = 3 biologically independent samples). Data are presented as mean ± SD. (one-way ANOVA and Tukey multiple comparisons tests). **f** The transendothelial electrical resistance (TEER, Ω cm$^{-2}$) values in the in vitro BBB model at different time points after incubation with HM-NPs, MM-NPs, CM-NPs or NPs. (*n* = 3 biologically independent samples). Data are presented as mean ± SD. (one-way ANOVA and Tukey multiple comparisons tests). **g** The TEER (Ω cm$^{-2}$) values in the

in vitro BBB model pretreated with cyclic adenosine monophosphate (cAMP) inhibitors (8-CPT-cAMP and Ro 20-1724) after incubation with HM-NPs, MM-NPs, CM-NPs or NPs. (*n* = 3 biologically independent samples). Data are presented as mean ± SD. **h** Schematic illustration showing the mechanism by which HM-NPs traverse the BBB. Western blot images of ZO-1 and claudin-5 expression in hCMEC/D3 cells during-treatment (**i**) and post-treatment (**j**) with HM-NPs, MM-NPs, CM-NPs, bare NPs or PBS. The immunoblots were representative data from three independent experiments. Immunofluorescence staining of ZO-1 in excised brain tissue from mice treated with HM-NPs, MM-NPs, CM-NPs, NPs or PBS during-treatment (**k**) and post-treatment (**l**). Red: ZO-1, Blue: Hoechst, scale bar = 20 μm. The immunofluorescence analyses were representative data from three mice. Source data are provided as a Source Data file.

slight increase for HM-NPs@G while a subsequent and sharp reduction for TMZ and PBS treatments, supporting the effective anti-GBM effects of HM-NPs@G (Fig. 6d). Importantly, the median survival time of mice treated with HM-NPs@G (65 d) was significantly longer in comparison to that of free TMZ and PBS group, which were 40 d and 19 d, respectively (Fig. 6e). The images of tumour-bearing brain slices stained with H&E showed that the tumour volume of HM-NPs@G group was the smallest (Fig. 6f). The TUNEL results indicated that more apoptotic GBM tumour cells were observed in the slices followed HM-NPs@G treatment (Fig. 6g). Taken together, the HM-NPs@G nanomedicines demonstrate enhanced therapeutic outcome than free TMZ, providing a potential alternative drug to be used in clinic.

### Routine blood and biochemical parameters

To assess safety, routine blood and biochemical tests were performed. No significant changes in the level of all blood parameters including alanine aminotransferase (ALT), aspartate aminotransferase (AST), alkaline phosphatase (ALP), plasma urea (BUN), uric acid (UA), creatinine (CR), blood platelet (PLT), red blood cells (RBCs) or white blood cell count (WBCs) were observed after treatment with HM-NPs@G, free Gboxin or PBS (Supplementary Figs. 24, 25). Furthermore, the levels of the pro-inflammatory cytokines IL-1β, IL-6 and Tnf-α in the liver and kidney of the mice following a single-dose of HM-NPs@G or free Gboxin were similar to that for PBS (Supplementary Fig. 25). Taken together with H&E staining of tissues taken from mice following multi-dose treatment, HM-NPs@G showed low systemic toxicity and good biocompatibility whereas kidney toxicity was caused by free Gboxin after single or multiple doses.

### Discussion

Glioblastoma is one of the deadliest cancers and features a poor prognosis. Gboxin as an oxidative phosphorylation (OXPHOS) inhibitor, can specifically suppress the growth of GBM cells with a notable low half-maximal inhibitory concentration (IC$_{50}$), however it suffers from poor BBB penetration and non-specific GBM targeting which limits its clinical potential. Mechanistically, Gboxin decreases the production of cellular energy (ATP) by inactivating ATP synthase which disrupts the electron transport chain, leading to decreased mitochondrial membrane potential and ultimately to structural damage. When mitochondrial structure is compromised and cytochrome c (Cyto C) is released to the cytosol which recruits and activates caspase proteins results in GBM cell apoptosis. Therefore, efficient BBB penetration, specific GBM and mitochondria targeting should boost the further clinical application of Gboxin.

To overcome the limitations of Gboxin treatment, we developed a cancer cell-mitochondria hybrid membrane camouflaged nanoplatform to efficiently pass BBB and specifically deliver Gboxin to the mitochondria of GBM cells. Hybrid membrane cloaked Gboxin-loaded biomimetic nanoparticles decreased nonspecific damage to normal cells (5–10 folds lower IC$_{50}$ than free Gboxin) but induced potent GBM tumour cell growth inhibition, especially in sensitive

GSCs. These results reflect the GBM tumour cell and mitochondria homologous dual-targeting, in addition to the use of ROS-responsive fast drug release polymeric core. Accordingly, our biomimetic nanomedicine addresses the key defects of Gboxin, i.e., poor pharmacokinetic profile by greatly extending circulation half-life (4.90 h versus 0.47 h of free Gboxin) by avoiding immuno-clearance. In turn, this greatly improves long-distance delivery and allows Gboxin to achieve maximal anti-tumour activity. Moreover, use of cancer cell-mitochondria hybrid membrane camouflaging enables NPs to exhibit excellent BBB penetration via down-regulating the proteins in tight junctions reducing junction tightness and allowing NPs to breach the BBB. As a result, these biomimetic nanoparticles achieve potent GBM tumour inhibition in vitro and in vivo leading to prolonged median survival time in U87MG and GBM stem cell X01 orthotopic mouse models. Importantly, negligible side effects were caused by HM-NPs@G, again reflecting the benefits arising from specific tumour cell and mitochondria dual-targeting and selective Gboxin release at high ROS levels present in disease lesions. It should be noted that PEG-PHB polymeric nanocarriers used for delivering Gboxin into the brain are inevitable and necessary due to the following points: (1) the polymeric nanocarriers could effectively encapsulate the Gboxin to protect it from degradation during the blood circulation, improving the extremely short circulation time of Gboxin. (2) the ROS-responsive PEG-PHB triggers the release of loaded Gboxin in tumour tissues and cells possess high level of ROS, while preventing the drug release in normal physiological conditions, leading to potent anticancer effects with few side effects. Therefore, it is indispensable to employ polymeric nanocarriers to deliver Gboxin in the GBM therapy. Though U87MG is not the perfect model to mimic the pathologies of GBM patients, there is no doubt to employ it as a GBM model in the initial proof-of-concept studies to verify the anti-tumour effect in preclinical studies[56-60]. Therefore, we firstly demonstrated the efficacy of HM-NPs@G towards the U87MG model, and the results strongly suggested that HM-NPs@G could inhibit tumour growth by activating mitochondria-related apoptosis. Furthermore, we adopted patient-derived GBM stem cells (GSCs) to establish mouse models and the results showed that the therapeutic outcomes are in line with that of U87MG model, further confirmed the superior effects of this biomimetic nanomedicine which is a potential formulation be translated in clinical for treating GBM.

More broadly, the combination of cancer cell and cell organelle hybrid membrane coating onto nanoparticles provides a promising membrane coating technology to achieve a wide range of bio interfacing functions for promoting specific and accurate subcellular targeted delivery. As cell organelles play a crucial role in modulating multiple functions within the cells, utilization of membranes of other cell organelles, such as the nucleus, Golgi apparatus, and lysosomes, have the potential to amplify cell biomimetic nanosystems. Looking towards the future, multifunctional nanomedicines with 'programmable' cell membranes have the opportunity to achieve a wide range of chemo/immunotherapy applications for many brain-related diseases.

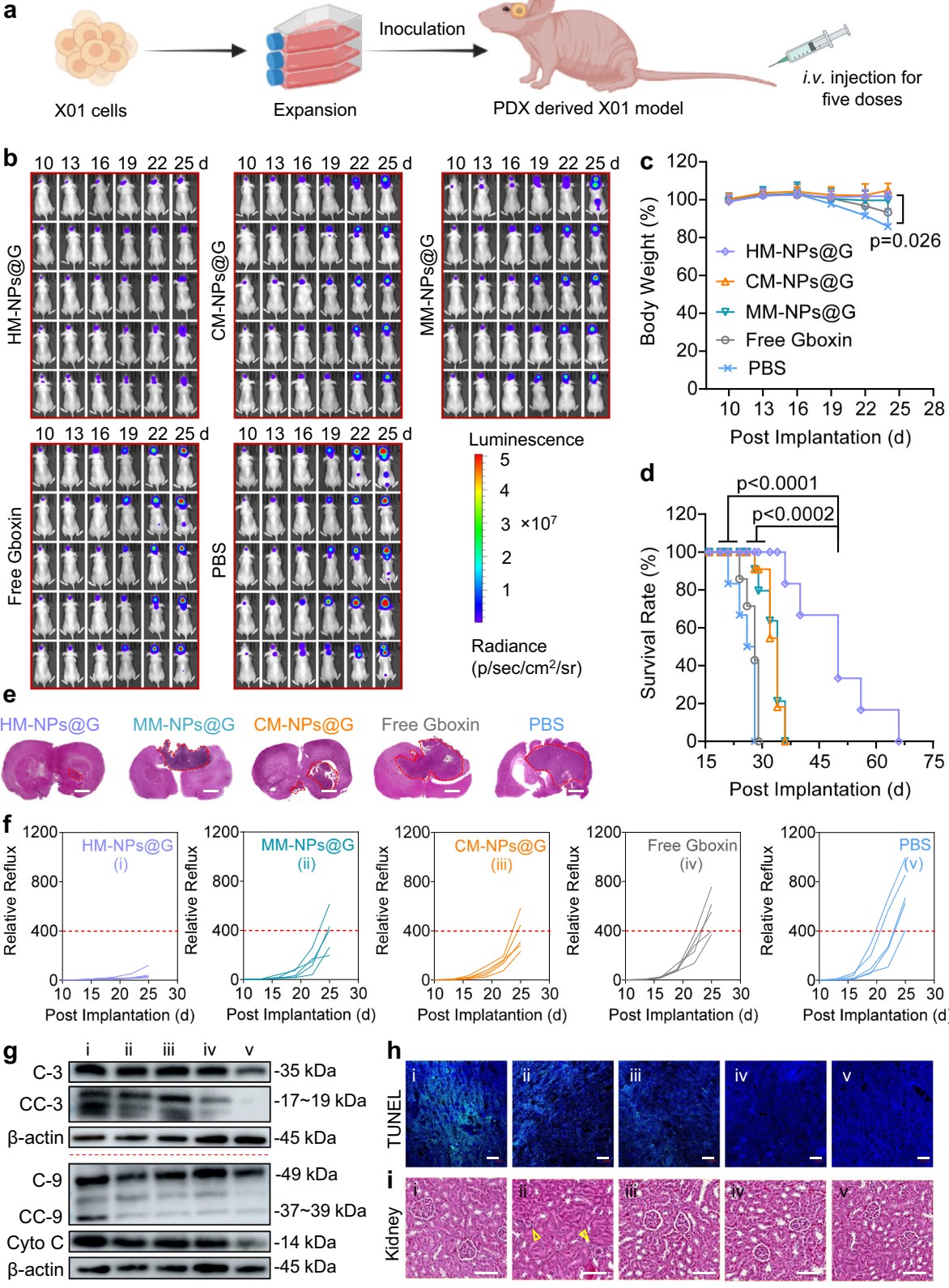

## Methods

### Ethical regulations

All animal handling protocols and experiments were approved by the Medical and Scientific Research Ethics Committee of Henan University School of Medicine (P. R. China) (HUSOM-2018-355). The mice must be euthanized once their weight loss reaches 20%.

### Materials

All chemicals were purchased from Sigma-Aldrich unless otherwise noted. Methoxypoly (Ethylene Glycol) 2000 Amine was synthesized by Jenkem Technology (Beijing, China). 4-(Hydroxymethyl) phenylboronic acid pinacol ester was purchased from Accela (Beijing, China). Gboxin and MFI8 were purchased from MedChemExpress

**Fig. 5 | In vivo antitumour activity of HM-NPs@G in mice bearing GBM stem cells (GSCs) xenografts. a** Schematic illustration of the establishment of the PDX-derived GSCs orthotopic model. **b** Time course luminescence images of mice bearing orthotopic X01-Luc GSC tumours following treatment with HM-NPs@G, MM-NPs@G, CM-NPs@G, free Gboxin or PBS ($n = 5$ mice in each group). The mice were intravenously injected at a dose of 3 mg Gboxin equiv. kg$^{-1}$ on day 10, 13, 16, 19 and 22 post tumour implantations. **c** Body weight profile ($n = 5$ mice in each group). Data are presented as mean ± SD (one-way ANOVA and Tukey multiple comparisons tests). **d** Mice survival rate curves. Statistical analysis: HM-NPs@G vs. MM-NPs@G or CM-NPs@G, *$p < 0.05$, HM-NPs@G vs. free Gboxin or PBS, **$p < 0.01$ ($n = 5$ mice in each group, Kaplan-Meier analysis, log-rank test). **e** H&E images of the whole brain excised from mice treated as described above on day 25 (Scale bar = 2 mm). The histological analyses were representative data from three mice. **f** Quantified luminescence levels of mice using the Lumina IVIS III system ($n = 5$ mice in each group). **g** Western blot analysis of apoptosis associated proteins and Cyto C in tumour tissues excised from the mice on day 25. The immunoblots were representative data from three independent experiments. **h** Histological analysis using TUNEL assay. Green: apoptotic cells; blue: Hoechst-stained cell nuclei (Scale bar = 50 μm). The histological analyses were representative data from three mice. **i** H&E images of the kidney excised from the mice treated with different nanoparticles as described above on day 25. The histological analyses were representative data from three mice. Source data are provided as a Source Data file.

(Shanghai, China). 8-(4-chlorophenylthio)-adenosine 3′,5′-cyclic monophosphate (8-CPT-cAMP, ab120424) were obtained from Abcam (Shanghai, China). Penicillin streptomycin, DMEM, fetal bovine serum (FBS) and 0.25% (w/v) trypsin solution were purchased from Gibco BRL (Gaithersberg, MD, USA). Membrane and cytosol protein extraction kit, cell mitochondria isolation kit, cell counting kit-8 (CCK-8), Annexin V-FITC apoptosis detection kit, one step TdT-mediated dUTP nick-end labeling (TUNEL) apoptosis assay kit, enhanced ATP assay kit, mitochondrial membrane potential assay kit with JC-1, 1,1′-dioctadecyl-3,3,3′,3′-tetramethylindodicarbocyanine,4-chlorobenzenesulfonate Salt (DiD), 3,3′-dioctadecyloxacarbocyanine perchlorate (DiO), mitotracker red, DAPI, Triton X-100, and bicinchoninic acid (BCA) test kit were provided by Beyotime Biotechnology Co., Ltd. (Nantong, China). Acetonitrile (HPLC grade) and methyl alcohol (HPLC grade) were purchased from Tianjin Siyou Co., Ltd. (Tianjin, China). 4-(3-Butoxy-4-methoxybenzyl)-2-imidazolidinone (Ro 20-1724, 29925-17-5) was purchased from Santa Cruz Biotechnology (USA). The oxygen consumption assay kit (BB-48211) and FCCP were the products of Bestbio (Shanghai, China). Antibodies used in western blotting: Anti-Cytochrome C rabbit antibody (Abcam, Catalog no.ab133504, 1/5000 dilution), Anti-Claudin 5 rabbit antibody (Abcam, Catalog no.ab131259, 1/5000 dilution), Caspase-9 Antibody (Human Specific) rabbit antibody (cell signaling technology, Catalog no.9502 S, 1/1000 dilution), Caspase-3 rabbit antibody (cell signaling technology, Catalog no.9662 S, 1/1000 dilution) and Cleaved Caspase-3 (Asp175) (5A1E) rabbit antibody (cell signaling technology, Catalog no.9664 S, 1/1000 dilution), ZO-1 rabbit Polyclonal antibody (Beyotime, Catalog no.AF8394, 1/1000 dilution), Anti-CD44 rabbit antibody (Abcam, Catalog no. ab243894, 1/1000 dilution), Anti-EpCAM rabbit antibody (Abcam, Catalog no. ab223582, 1/1000 dilution), Anti-EHD2 rabbit antibody (Abcam, Catalog no. ab154784, 1/5000 dilution), Anti-Mitofusin rabbit antibody (Abcam, Catalog no. ab221661, 1/1000 dilution), Atlastin-1 rabbit antibody (cell signaling technology, Catalog no. 12728 S, 1/1000 dilution), Integrin αV rabbit antibody (cell signaling technology, Catalog no. 60896 S, 1/1000 dilution), Na,K-ATPase rabbit antibody (cell signaling technology, Catalog no. 3010 S, 1/1000 dilution), β-actin rabbit antibody (Thermofisher, Catalog no.BS-50545R, 1/5000 dilution), HRP Goat Anti-Rabbit IgG (H&L) (UElandy, Catalog no. H6162S/H6162, 1/25000 dilution), HRP Goat Anti-Mouse IgG (H&L) (UElandy, Catalog no. H6161S/H6161, 1/25000 dilution). Antibodies used in immunohistochemistry staining: Ki67 rabbit polyclonal antibody (Servicebio, Catalog no.GB111499, 1:500 dilution), Cleaved Caspase 3 rabbit polyclonal antibody (Servicebio, Catalog no.GB11532, 1:500 dilution). Antibodies used in immunofluorescence analysis: Anti-hsp60 rabbit polyclonal antibody (Servicebio, Catalog no.GB11243, 1/1000 dilution), Anti-Nephrin rabbit polyclonal antibody (Servicebio, Catalog no. GB11343, 1/1000 dilution).

## Cell lines
The human GBM U87MG cell line, human brain endothelial hCMEC/D3 cell line, N2a cell line, BV2 cell line, U251 cell line, U251TR cell line were purchased from the the American Type Culture Collection (ATCC). U87MG-Luc cell line was purchased from Shanghai Model Organisms Center (Shanghai, China). The above cell lines were cultured in DMEM medium supplemented with 10% (V/V) fetal bovine serum and 1% (V/V) penicillin and streptomycin. The X01 glioblastoma stem cells (GSCs) were kindly provided by Professor Jong Bae Park from the National Cancer Center of South Korea, the more details are in the previous works[49,61–64]. X01 cells were maintained in DME/F12 supplemented with epidermal growth factor (EGF, 10 ng ml$^{-1}$), basic fibroblast growth factor (bFGF, 10 ng ml$^{-1}$), B27, and 1% penicillin and streptomycin. These cells were cultured as a monolayer in a humidified atmosphere containing 5% $CO_2$ at 37 °C.

## Animal models
BALB/c mice and nude mice (female, 6–8 weeks) were purchased from Sipeifu (SPF) biotechnology (Beijing, China). Mice were feed under a 12 h light-dark cycle, 20–24 °C and 45–65% relative humidity. An orthotopic U87MG glioblastoma (GBM) bearing mouse model was established with a high success rate of nearly 100% via implantation of minced glioblastoma tissue into the left striatum of BALB/c nude mice as our previous work[27]. The growth of the GBM was monitored by bioluminescence using an imaging system (IVIS lumina III, Perkinelmer, USA), 10 min after the mice were anesthetized combined with luciferase substrate D-luciferin potassium (15 mg mL$^{-1}$ dissolved in PBS) at 150 mg kg$^{-1}$.

## Synthesis and characterization of 4-(4, 4, 5, 5-Tetramethyl-1, 3, 2-dioxaborolan-2-yl) benzyl acrylate (HB) monomer
4-(Hydroxymethyl) phenylboronic acid pinacol ester (S2) (2.925 g, 12.5 mmol) was dissolved in anhydrous dichloromethane (DCM, 15 mL), followed by adding 2.09 g triethylamine (TEA,12.5 mmol). After cooling to -0 °C, 1.262 g (15 mmol) of methyl acryloyl chloride in -1.25 mL dried DCM was added dropwisely within 1 h. Then, the reaction mixture was stirred for 10 h at room temperature followed by filtering. The filtrate was concentrated on a rotary evaporator and diluted by ethyl acetate, and washed with brine thrice. After drying with $MgSO_4$ overnight, the organic solution was concentrated and purified by silica column chromatography using petroleum ether and ethyl acetate (v/v = 30/1) as the eluent. The final product was obtained as yellow oil with the yield of 35%. $^1$H NMR spectrum of HB monomer was recorded with Bruker Avance III HD 400 MHz spectrometer.

## Synthesis of PEG-CPADN macroinitiator
CPADN (125.7 mg, 0.45 mmol) was dissolved in anhydrous tetrahydrofuran (THF, 5 mL), followed by adding 56.96 mg NHS (0.495 mmol). After cooling to -0 °C, 102.13 mg (0.495 mmol) of DCC in 5 mL dried THF was added dropwisely within 1 h. Then, the reaction mixture was stirred for 24 h at room temperature, followed by adding 200 μL TEA. 200 mg poly (ethylene glycol) with the end group of amine (PEG, 2 K) was added into the mixture for 12 h reaction. After that, the reaction mixture was filtered, the filtrate was concentrated on a rotary evaporator and precipitated into cold diethyl ether. The precipitate was then dissolved in DCM and repeated the

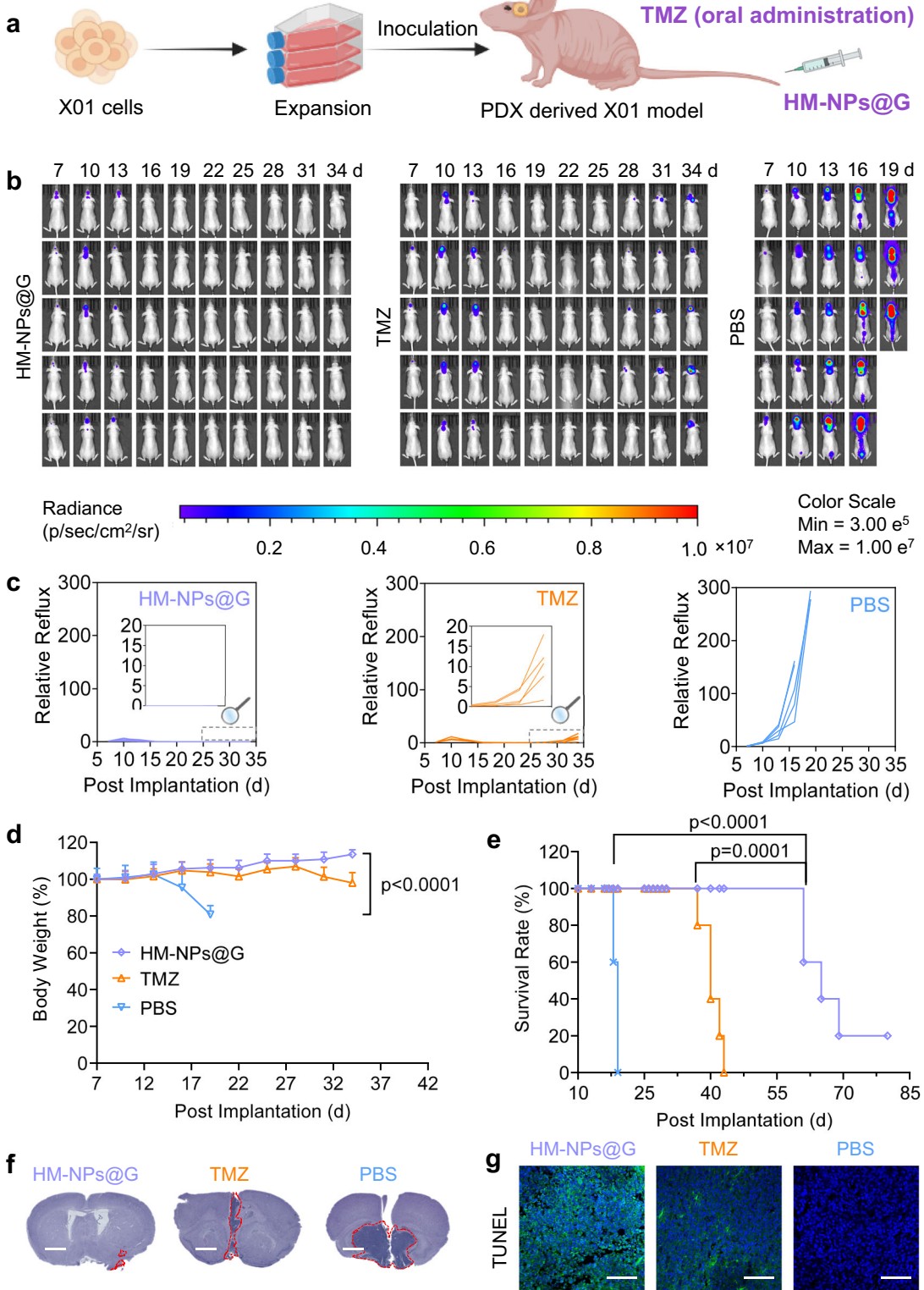

**Fig. 6 | In vivo antitumour activity of HM-NPs@G in mice bearing GBM stem cells (GSCs) xenografts. a** Schematic illustration of the establishment of the PDX-derived GBM GSCs orthotopic model. **b** Time course luminescence images of mice bearing orthotopic X01-Luc tumours following treatment with HM-NPs@G, free TMZ or PBS ($n = 5$ mice in each group). The mice were intravenously injected with HM-NPs@G or orally injected with TMZ (the dosage for both the Gboxin and TMZ were 5 mg equiv. kg$^{-1}$) on day 7, 10, 13, 16 and 19 post tumour implantation. **c** Quantified luminescence levels of mice using the Lumina IVIS III system ($n = 5$ mice in each group). **d** Body weight profile ($n = 5$ mice in each group). Data are

presented as mean ± SD (one-way ANOVA and Tukey multiple comparisons tests). **e** Mice survival rate curves. Statistical analysis: HM-NPs@G vs. TMZ, \*\*$p < 0.01$, HM-NPs@G vs. PBS, \*\*\*$p < 0.001$. ($n = 5$ mice in each group, Kaplan-Meier analysis, log-rank test). **f** H&E images of the whole brain excised from mice treated as described above on day 22 (Scale bar = 2 mm). The histological analyses were representative data from three mice. **g** Histological analysis using TUNEL assay. Green: apoptotic cells; blue: Hoechst-stained cell nuclei (Scale bar = 100 μm). The histological analyses were representative data from three mice. Source data are provided as a Source Data file.

dissolution-precipitation process twice. The final product was dried in a vacuum oven, obtaining a pink solid with the yield of 72%. $^1$H NMR spectrum of PEG-CPADN was recorded with Bruker Avance III HD 400 MHz spectrometer.

## Synthesis and characterization of poly (4-(4, 4, 5, 5-tetramethyl-1, 3, 2-dioxaborolan-2-yl) benzyl acrylate) (PEG-PHB) block polymer

The PEG-PHB was prepared via reversible addition-fragmentation chain transfer (RAFT) polymerization of HB monomer using the above synthesized PEG-CPADN as the chain transfer agent. Briefly, PEG-RAFT (0.01 mmol, 25 mg), AIBN (0.002 mmol, 0.328 mg), HB monomer (0.2 mmol, 60.8 mg) was dissolved in 2 mL 1, 4-dioxane and reacted in a 5 mL Schlenk flask. The reaction system was degassed by three freeze-pump-thaw cycles and sealed under vacuum. Then, the Schlenk flask was placed in a preheated oil bath at 65 °C. After 48 h, the impurities in this reaction mixture were removed by dialysis against ddH$_2$O and lyophilized to obtain the final block polymer with the yield of 70%. $^1$H NMR spectrum of PEG-PHB was recorded with Bruker Avance III HD 400 MHz spectrometer.

## Preparation of cancer cell and mitochondria membrane fragments

The human GBM U87 MG cell membrane was obtained using a membrane protein extraction kit. Briefly, the collected cells were dispersed in membrane protein extraction buffer solutions and cooled in an ice bath for 10–15 min. After that, the cells were subjected to 3 cycles of freezing-thawing. Then, the obtained mixture was centrifuged (700 g, 10 min, 4 °C), and the supernatant was further centrifuged (14,000 g, 30 min, 4 °C). Finally, the cancer cell membrane (CM) was obtained by lyophilizing the precipitate.

Mitochondria membrane (MM) was prepared according to the mitochondria extraction kit first followed by splitting the mitochondria. The cells were collected by centrifugation at 700 g for 3 min. The collected cells were washed with PBS and suspended in mitochondria extraction reagent A containing PMSF (1 mM). The mixture was incubated in an ice bath for 15 min and then homogenized 20 times by a suitable size glass homogenizer to break up the cells. Next, the cell homogenate was centrifuged at 4 °C, 600 g for 10 min. The supernatant was centrifuged at 4 °C, 11,000 g for 10 min to obtain mitochondria. The mitochondria membrane was acquired via breaking up by lysis buffer and ultracentrifuging at 4 °C, 100,000 g for 70 min to obtain the MM, which was lyophilized and stored at −80 °C for further use.

## Preparation and characterization of hybrid membrane

Förster resonance energy transfer (FRET) was employed to study the fusion process. Briefly, 1 mL MM solution and 10 μL 1, 1'-dioctadecyl-3, 3, 3', 3'-tetramethylindocarbocyanine perchlorate (DiO) (excitation/emission = 484/501 nm) solution (5 μg/μL) were mixed and dyed for 20 min under darkness, 1 mL CM solution and 10 μL (DiD) (excitation/emission = 644/665 nm) (5 μg/μL) solution were mixed and dyed for 20 min without light. After that, the membrane solution was centrifuged at 100,000 g and 21,000 g for 60 and 30 min at 4 °C, respectively. Then washed with PBS three times to remove excess dyes. The resulting membrane fractions were resuspended in 1 mL PBS solution. The MM solution was added to the CM solution at the membrane protein weight ratio of 1: 1 (MM: CM). The sample was sonicated in an ice bath for 2 min and subsequently extruded through 800 nm, 400 nm, and 200 nm polycarbonate porous membranes using an Avanti mini extruder to facilitate membrane fusion. Finally, the hybrid membrane vesicles (HM) were collected by centrifugation (21,000 g, 30 min, 4 °C) and re-suspended in PBS. The HM was further characterized by confocal laser scanning microscopy (CLSM).

## Characterization of cancer membrane and mitochondria by western blotting

The obtained cancer membrane (U87MG), GBM stem cell (X01) membrane, mitochondria membrane (MM), NPs, HM-NPs (U87MG) and HM-NPs (X01) were lysed with radio immunoprecipitation assay (RIPA) lysis buffer at 4 °C for 10 min. The lysates were subjected to SDS-PAGE and then transferred onto PVDF membranes (Millipore). After being blocked in 5% skim milk for 1 h, the membranes were separately incubated with rabbit antibodies against CD44, EpCAM, EHD2, Atlastin-1, and Mito-fusion, Integrin αv and Na$^+$/K$^+$ATPase at 4 °C overnight, respectively, and then treated with anti-rabbit secondary antibody (1: 10,000) for 1 h at room temperature. Subsequently, the immunoreactive bands were visualized by enhanced chemiluminescence (Amersham Imager 680RGB, GE, Japan). Na$^+$/K$^+$ATPase was detected as a housekeeping protein control.

## Preparation and characterization of HM-NPs@G

Gboxin loaded nanoparticles (NPs@G) were prepared by solvent exchange self-assemble method. Briefly, PEG-PHB (1 mg) was dissolved in THF (200 μL), mixed with Gboxin (10 μL, 25 mg mL$^{-1}$) was dropped into HEPES buffer (1 mL, 10 mM), then stirred for 3 h and dialyzed (Spectra/Por; molecular weight cutoff [MWCO] 3500) for 4 h against HEPES to remove the unloaded Gboxin. The size and zeta potential of nanomedicine were determined at 25 °C using dynamic light scattering (DLS, Nano-Zen 3600, Malvern Instruments, UK). The measurements were carried out in triplicate. The concentration of Gboxin was detected by High Performance Liquid Chromatography (HPLC, Agilent G7129C). The analysis was performed on a Waters system with A: ddH$_2$O (0.01% TFA) and B: acetonitrile (0.01% TFA) as the eluent (5% to 95% B within 1.3 min, 95% to 5% B from 1.3 min to 3.0 min), flow rate: 1.8 mL min$^{-1}$, UV wavelength: 214/254 nm, injection volume: 10 μL, Column: SunFire C18 (50 mm × 4.6 mm, 3.5 μm), retention time: 1.4 min. DLC and DLE were obtained using the following formulae:

$$DLC (wt\%) = (weight\ of\ loaded\ drug/total\ weight\ of\ the\ polymer\ and\ loaded\ drug) \times 100$$

$$DLE (\%) = (weight\ of\ loaded\ drug/weight\ of\ the\ drug\ in\ feed) \times 100$$

Next, to fabricate the hybrid membrane-coated nanoparticles (HM-NPs@G), the NPs@G solution (1 mL, 1 mg mL$^{-1}$) was added to the HM suspension (1 mL, 1 mg mL$^{-1}$) and vortex stirring, then extruded consecutively through a series of water-phase filters with reducing pore sizes (800 nm and 400 nm). SDS-PAGE was employed to examine the protein profile of cancer membrane, mitochondria membrane, hybrid membrane and hybrid membrane coated nanoparticles (20 mg of protein sample was loaded in each lane, constant current: 50 mA for 1 h).

## Characterization of HM-NPs@G by TEM

The structure of the nanomedicine was examined using a transmission electron microscope (TEM). 10 μL of the nanomedicine solution was deposited onto a glow-discharged carbon-coated grid. After 10 min, the grid was washed with 10 drops of distilled water. A drop of 1% uranyl acetate stain was added to the grid. Dyeing for another 10 min, the stain was washed with distilled water. The grid was subsequently dried and visualized using TEM (JEM-2010HT, Japan).

## Calculation details

All the calculations were performed with the Gaussian 16 program package[65] using the default conditions implemented in it. For geometry optimization and frequency analysis, we adopted the B3LYP-D3BJ[66] with the 6−31 G (d,p)[41,67,68] basis set. Optimized minima were proved by vibrational analysis to have no frequency. Moreover, Basis Set Superposition Error (BSSE) was also considered.

### In vitro ROS-responsive of HM-NPs@G

To determine the responsiveness of HM-NPs@G when exposed to reactive oxygen species (ROS), the in vitro release kinetic of Gboxin from HM-NPs@G nanomedicines was evaluated using a dialysis tube (MWCO 12,000–14,000) under shaking (37 °C, 200 rpm) in PB (pH 7.4, 10 mM) with or without $H_2O_2$ (0.1 mM $H_2O_2$, 1 mM $H_2O_2$). Typically, HM-NPs@G was dialyzed against 25 mL of release media. At each time interval, the solution outside the dialysis membrane (5 mL) was withdrawn and replaced with the same volume of fresh media. The released Gboxin was determined by HPLC as above. The release experiments were conducted in triplicate and the results presented were the average data with standard deviations. The size and PDI changes of HM-NPs@G when exposed to $H_2O_2$ were monitored with DLS.

### Uptake analysis by flow cytometry

To assess the homotypic targeting effect of membrane-coated NPs, U87MG cells were seeded in 12-well plates ($5 \times 10^5$ cells/well). After incubation for 24 h, the cells were treated with Cy5 loaded HM-NPs, MM-NPs, CM-NPs, NPs (Cy5: 10 μg mL$^{-1}$) and incubated for 6 h. PBS-treated cells were used as control. The cells were washed three times with cold PBS, harvested and recorded immediately using a flow cytometer (Becton Dickinson, USA), then analyzed using Cell Quest software based on 10,000 gated events. The gate was arbitrarily set for the detection of Cy5 fluorescence.

### Cellular uptake assay by CLSM

The cellular uptake behavior of HM-NPs was detected with CLSM in U87MG cells. The cells were cultured on microscope slides in a 6-well plate ($1 \times 10^5$ cells/well) and further were incubated with HM-NPs, CM-NPs, MM-NPs, NPs (FITC concentration: 10 μg mL$^{-1}$) for 6 h. Then the culture medium was removed, the cells on microscope plates were washed with PBS three times, stained with MitoTracker (red) for 20 min, fixed with 4% paraformaldehyde solution for 15 min and stained with Hoechst (blue) for 10 min and finally mounted with glycerol. The fluorescence images were obtained using a confocal microscope (Zeiss LSM 880, Germany).

### Co-localization of HM-NPs with mitochondria detected by Bio-TEM

U87MG cells were grown in 6-well plate ($5 \times 10^4$ cells/well). After 24 h, the cells were incubated with HM-UCNPs, CM-UCNPs, MM-UCNPs and UCNPs at 37 °C for 72 h (UCNPs: 1 mg mL$^{-1}$). Then the culture medium was removed, the cells were pre-fixed using 2.5% glutaraldehyde for 5 min at room temperature. Cells were further fixed with 2.5% glutaraldehyde for 30 min at room temperature, washed three times with PBS and dehydrated in an ascending gradual series of ethanol (50%, 70%, 90%, 95%, and 100%) for 8 min. Samples were infiltrated with and embedded in SPON12 resin. After polymerizing for 48 h at 60 °C, 70 nm-thick ultrathin sections were cut using a diamond knife, and then picked up with Formvar-coated copper grids (100 mesh). The sections were double-stained with uranyl acetate and lead citrate. After air drying, samples were examined with TEM.

### Mitochondria targeting mechanism investigation of HM-NPs in vitro

U87MG cells were incubated in 6 cm plates ($2 \times 10^5$ cells/well) overnight and then pre-treated with MFI8, a small inhibitor of mitofusin protein, for 6 h (MFI8: 5 μM). HM-NPs@G were added following the MFI8 pretreatment and incubated for further 24 h (Gboxin: 20 μg mL$^{-1}$). After that, the cells were collected and counted. Then, the mitochondria of the same amount of U87MG cells were isolated according to the mitochondria extraction kit. The isolated mitochondria were lysed and the Gboxin accumulation in mitochondria was detected with HPLC. The accumulation of HM-NPs in mitochondria without MFI8 preprocessing was used as control.

### The cytotoxicity of HM-NPs@G by CCK-8 assay

The U87MG or GBM stem cells (GSCs) X01 were plated in a 96-well plate ($1 \times 10^3$ cells/well) for 24 h. The culture medium was removed and replenished with 100 μL fresh medium containing HM-NPs@G, CM-NPs@G, MM-NPs@G, NPs@G and free Gboxin were added to yield a final concentration of Gboxin as 800 nM. After 72 h incubation, the medium was replaced by 100 μL fresh medium containing 10 μL of CCK-8 solution (5 mg mL$^{-1}$). After incubation for another 40 min or 4 h, the absorbance at a wavelength of 450 nm of each well was measured using a microplate reader (Devivces/13x, Molecular Device, USA). Cells treated with PBS were used as controls.

### IC$_{50}$ assay of Gboxin and HM-NPs@G to various cells

The normal cells including HA1800, hCMEC/D3, N2a and BV2 cells, GBM cells (U87MG, U251 and U251-TR) as well as GSCs (X01) were plated in a 96-well plate ($2 \times 10^3$ cells/well) using medium supplemented with 10% fetal bovine serum (FBS) for 24 h. The culture medium was removed and replenished with free Gboxin and HM-NPs@G at different concentrations. The cells were cultured at 37 °C in an atmosphere containing 5% $CO_2$ for 72 h. The medium was replaced by 90 μL of fresh medium. 10 μL of CCK solution (5 mg/mL) was added. The cells were incubated for another 40 min or 4 h. The absorbance at a wavelength of 450 nm of each well was measured using a microplate reader.

### Apoptosis assay of HM-NPs@G

The cell apoptosis effect of HM-NPs@G was measured using Annexin V-FITC/PI apoptosis detection kit. U87MG or X01 cells ($1 \times 10^5$ cells/well) were seeded in the 12-well plates for 24 h and cultured with HM-NPs@G, CM-NPs@G, MM-NPs@G, NPs@G and free Gboxin (Gboxin: 800 nM). After 72 h incubation, the U87MG or X01 cells were washed 3 times with PBS, and stained with 10 μL iodide and 5 μL Annexin V-FITC, cultured for 15 min in the dark. The cells were resuspended in 300 μL PBS followed by analyzing with a flow cytometer.

### Oxygen consumption rate (OCR) assay of HM-NPs@G

OCR was measured according to the protocol of BBoxiProbe™ R01 kit. Briefly, U87MG or X01 cells were plated in a transparent bottom and black side 96-well plates ($8 \times 10^4$ or $6 \times 10^4$ cells/well) and incubated overnight. Next, 100 μL fresh medias containing HM-NPs@G, CM-NPs@G, MM-NPs@G and free Gboxin (Gboxin: 800 nM) were added to each well, and 4 μL BBoxiProbe®RO1 oxygen fluorescence probe was added and mixed fully with the media. Meanwhile, 100 μL of oxygen blocking buffer was promptly added. Then, the fluorescence at an excitation wavelength of 455–468 nm and an emission wavelength of 603 nm of each well was measured using a microplate reader (Devivces/13x, Molecular Device, USA) at 3-min intervals until unchanging within 2 h. Cells treated with PBS were used as controls. The oxygen consumption rate (%) = (final fluorescence in cells treated different drugs – initial fluorescence in cells treated with corresponding drug) / (final fluorescence in control cells – initial fluorescence in control cells) × 100%.

### ATP activity detection

U87MG or X01 cells were plated in 24-well plates ($1 \times 10^5$ cells/well) and incubated overnight. Following 72 h incubation in fresh media containing HM-NPs@G, CM-NPs@G, MM-NPs@G and free Gboxin (Gboxin: 800 nM), the cells were washed twice with PBS and lysed with ATP lysis buffer. The intracellular ATP levels were determined following the protocol of the ATP assay kit (Beyotime Institute of Biotechnology, China), then measured using a microplate reader by calibration with the ATP standards.

### Mitochondrial membrane potential analysis

The U87MG cells were cultured on microscope slides in a 12-well plate ($1 \times 10^6$ cells/well) and added 10 μL HM-NPs@G, CM-NPs@G, MM-NPs@G and free Gboxin (Gboxin: 800 nM). After 72 h

incubation, the cells with different treatments were stained with JC-1 probe and detected by CLSM as above. Red fluorescence represents the aggregate form of JC-1, indicating high mitochondrial membrane potential (ΔΨm) while the green fluorescence represents the monomeric form of JC-1, indicating low ΔΨm.

## Mitochondria structure damage detected by Bio-TEM

U87MG cells were grown in 10 mm dishes. After 12 h, the cells were incubated with free Gboxin, NPs@G, CM-NPs@G, MM-NPs@G, and HM-NPs@G at 37 °C for 72 h. Then the culture medium was removed, the cells were pre-fixed using 2.5% glutaraldehyde for 5 min at room temperature. Cells were further fixed with 2.5% glutaraldehyde for 30 min at room temperature, washed three times with PBS and dehydrated in an ascending gradual series of ethanol (50%, 70%, 90%, 95%, and 100%) for 8 min. Samples were infiltrated and embedded in SPON12 resin. After polymerizing for 48 h at 60 °C, 70-nm-thick ultrathin sections were cut using a diamond knife, and then picked up with Formvar-coated copper grids (100 mesh). The sections were double-stained with uranyl acetate and lead citrate. After air drying, samples were examined with TEM. HA1800 microglia cell sample was prepared and observed similarly as a control.

## Western blotting

U87MG cells were incubated in 6-well plates ($2 \times 10^5$ cells/well) for 24 h, then added with 10 μL HM-NPs@G, CM-NPs@G, MM-NPs@G and free Gboxin (Gboxin: 800 nM), respectively. After 72 h incubation, the cells were washed three times with PBS and lysed with radio RIPA lysis buffer at 4 °C for 10 min. The lysates were subjected to SDS-PAGE and then transferred onto PVDF membranes (Millipore). After being blocked in 5% skim milk for 1 h, the membranes were separately incubated with rabbit antibodies against caspase-3/9 (C-3/9), cleaved caspase-3/9 (CC-3/9), cytochrome C (Cyto C) and β-actin at 4 °C overnight, respectively, and then treated with anti-rabbit secondary antibody (1: 10,000) for 1 h at room temperature. Subsequently, the immunoreactive bands were visualized by enhanced chemiluminescence (Amersham Imager 680RGB, GE, Japan). β-actin was detected as a housekeeping protein control.

## On-targets of HM-NPs@G assay in vitro

U87MG cells were incubated in 6 cm plates ($2 \times 10^5$ cells/well) for 24 h, then added with 500 μL HM-NPs@G, CM-NPs@G, MM-NPs@G and free Gboxin (Gboxin: 20 μg mL$^{-1}$), respectively. After 24 h incubation, the cells were collected and counted. Then, the mitochondria of the same amount of U87MG cells were isolated according to the mitochondria extraction kit. The isolated mitochondria were lysed and the concentration of Gboxin was detected with HPLC.

## Penetration in U87MG multicellular spheroids

To observe the penetration abilities of HM-NPs in multicellular spheroids, the U87MG cells were seeded in a 96-well plate (Prime-Surface TM, MS-9096U) ($5 \times 10^3$ cells/well). After 48 h when the diameter of multicellular spheroids reached about 500 μm, the spheroids were cultured with Cy5 loaded HM-NPs, MM-NPs, CM-NPs and NPs for 6 h. The pellets were washed three times with PBS and then transferred into confocal dishes. The fluorescence of Cy5 was detected using Z-stack imaging, with 10 μm intervals from the bottom of the spheroids to the middle, using CLSM.

## BBB permeability evaluation in vitro BBB transwell model

The in vitro BBB model was established with endothelial cells (hCMEC/D3 cells) and astrocytes (HA1800 cells) using a transwell cell culture system. Briefly, HA1800 cells were seeded ($2 \times 10^5$/well) on the underside of a transwell chamber, and they were allowed to adhere for 24 h. Then, the transwell chamber was carefully turned up-right and hCMEC/D3 cells ($1 \times 10^5$/well) were seeded on the top of the Transwell chamber. The BBB penetration assay was conducted after HA1800 and

hCMEC/D3 cells were co-cultured to a high density. The integrity of the cell bilayer was evaluated by measuring the trans-epithelial electrical resistance (TEER) values using a Millicell-ERS voltohmmeter (Millipore, USA). The cell monolayers with TEER values higher than 200 Ω cm$^{-2}$ were used as the BBB model for the transmigration evaluation. Cy5 loaded HM-NPs, CM-NPs, MM-NPs and bare NPs (Cy5 concentration: 10 μg mL$^{-1}$) were added to the upper chamber, and the FBS-free medium was added to the lower chamber. After incubation for 2 h, 10 h and 24 h, the Cy5 fluorescence of supernatant in the upper chamber and the medium in the lower chamber were analyzed using a microplate reader. The transport ratio in each compartment was calculated according to the initial feeding amount of Cy5 loaded nanoparticles.

## BBB penetration mechanism investigation of glioblastoma cell membrane camouflaged nanoparticles

The HA1800 and hCMEC/D3 cells ($2 \times 10^4$ cells/filter) were planted on transwell filters (0.4 μm pore polycarbonate membrane inserts) and incubated to reach full confluency. After that, membrane coated nanoparticles as well as nanoparticles along with 8-(4-chlorophenylthio)-adenosine 3′,5′-cyclic monophosphate (8-CPT-cAMP, 50 nM) and 4-(3-butoxy-4-methoxybenzyl)-2-imidazolidinone (Ro 20-1724; 17.5 nM) were added into the upper chamber. To determine the integrity of the endothelial monolayer, the TEER was monitored at 0, 2, 4, 6, 8, 12, 24, 48 h. Additionally, to evaluate the BBB transport and uptake of membrane decorated nanoparticles towards U87MG cells, the fluorescence intensity was measured with flow cytometry after 48 h incubation.

The hCMEC/D3 endothelial cells were seeded in 6-well plates, followed by incubation for several hours until the tight junctions formed. Thereafter, the media was replaced with NPs, CM-NPs, MM-NPs and HM-NPs. After incubation for 48 h and 72 h respectively, the hCMEC/D3 cells were washed and collected for further ZO-1 and claudin-5 levels analysis by western blotting.

## BBB penetration mechanism verification in vivo with immunofluorescence

Paraffin-embedded tumors excised from U87MG cell tumor-bearing mice treated with HM-NPs, MM-NPs, CM-NPs, NPs, and PBS for 48 h and 72 h were cut into 5 mm-thick sections, deparaffinized in a xylene series, and hydrated in distilled water. Antigen retrieval was undertaken with citrate buffer and washing with PBS, followed by incubation with primary antibodies overnight at 4 °C. U87MG tumor-bearing brains frozen sliced were also stained with Hoechst. Finally, the stained slices were observed with CLSM.

## In vivo pharmacokinetics

Tumor-free BALB/c healthy mice were used for pharmacokinetic study of HM-NPs@G. 10–50 μL of blood was taken from the retro-orbital sinus of mice at different time points post-injection of HM-NPs@G, MM-NPs@G, CM-NPs@G, NPs@G or free Gboxin (5 mg Gboxin equiv. kg$^{-1}$). Each blood sample was immediately weighed and dissolved in 0.1 mL Triton X-100 with brief sonication. Subsequently, 0.2 mL acetonitrile was added to each blood sample and incubated at room temperature overnight. Samples were then vortexed and centrifuged at 8000 g for 15 min. The Gboxin content in the supernatant was determined by HPLC as described above.

## In vivo imaging of HM-NPs

The luciferase expressed U87MG (U87MG-Luc) human GBM orthotopic xenografts were randomly grouped and injected with Cy5 loaded HM-NPs, CM-NPs, MM-NPs, NPs and free Cy5 (Cy5 dosage: 2 mg kg$^{-1}$) in 200 μL PBS via tail vein. At predetermined time points (0, 2, 4, 6, 8, 12, and 24 h) post intravenous (i.v.) injection, the mice were anesthetized with 3% isoflurane and during the imaging acquisition process, 1% isoflurane anesthesia was delivered via a nose cone system. Fluorescent images were acquired using a near-infrared fluorescence

imaging system IVIS Lumia III at excitation of 647 nm and emission of 670 nm, and the images were analyzed using Lumia III software with the same fluorescence scale.

### Ex vivo imaging and tumor penetration

For ex vivo imaging and the tumor penetration behavior of HM-NPs were studied by immunofluorescent analysis. Briefly, Cy5 loaded HM-NPs, CM-NPs, MM-NPs, NPs and free Cy5 (Cy5 dosage: 1 mg kg$^{-1}$) were i.v. injected into U87MG orthotopic xenografts via tail vein. The mice were sacrificed at 6 h post-injection after perfusion, and major organs including heart, liver, spleen, lung, kidney and brain were taken. For tumor penetration, the fluorescence images of major organs were acquired with the Lumina IVIS III near-infrared fluorescence imaging system, and the GBM brain frozen sliced for immunofluorescent staining with mitochondria (Anti-hsp60 rabbit pAb, green) and cell nucleus (DAPI, blue). Finally, the stained slices were observed with CLSM.

### Biodistribution of HM-NPs

For biodistribution, HM-NPs@G, MM-NPs@G, CM-NPs@G, and free Gboxin in 200 μL HEPES (5 mg Gboxin equiv. kg$^{-1}$) were administrated i.v. via the tail vein into U87MG-Luc orthotopic tumor-bearing nude mice. At 6 h post injection, the mice were sacrificed and the major organs including heart, liver, spleen, lung, kidney, cancerous-brain were collected after perfusion, washed, and weighed. To quantify the Gboxin, the tumor block and major organs were homogenized in 0.6 mL of 1% Triton X-100 with a homogenizer at the frequency of 70k Hz for 6 min. After that, 200 μL acetonitrile was added and all samples were centrifuged at 8000 g for 30 min. The content of Gboxin in the supernatant was determined by HPLC based on a calibration curve as above.

Renal distribution of HM-NPs was studied by immunofluorescence stain. Tumor-free BALB/c healthy mice (6–8 weeks) received i.v. injections of HM-NPs@Cy5 (Cy5: 2 mg kg$^{-1}$). Then, these mice were sacrificed at three consecutive time points after injection (1 h, 6 h, and 48 h), and their kidney were collected. The kidney frozen slices were immunofluorescent staining with renal glomerulus (Anti-Nephrin Rabbit pAb, green) and cell nucleus (DAPI, blue). Finally, the stained slices were observed with CLSM.

### In vivo anticancer effect in GBM xenograft model

U87MG orthotopic GBM bearing mice were randomly divided into five groups ($n = 10$, 6 for monitoring survival, 4 for histological analysis and western blotting) at day 7 post-implantation, followed by receiving an i.v. injection of HM-NPs@G, CM-NPs@G, MM-NPs@G, and free Gboxin (3 mg Gboxin equiv. kg$^{-1}$) or PBS every three days for totally five doses. After the injection, the mice were anesthetized and their tumor luminescence intensity was monitored by the Lumina IVIS III system. The relative photon flux was normalized to initial intensity, $I/I_0$ ($I_0$ is the bioluminescence intensity at day 7). On day 22, the treatment was terminated and four mice from each group were sacrificed for histological analysis and western blot analysis. The body weight of mice was individually measured every three days and the Kaplan-Meier survival curve was recorded during the period.

To further investigate the apoptosis of tumor cells induced by Gboxin, brain tumor tissues were further stained with terminal deoxynucleotidyl transferase dUTP nick end labeling (TUNEL) and finally evaluated with a CLSM. In addition, the brain tumors from mice treated with different nanomedicines were lysed with RIPA lysis buffer at 4 °C for 10 min. The lysates were subjected to SDS-PAGE and then transferred onto PVDF membranes (Millipore). After being blocked in 5% skim milk for 1 h, the membranes were separately incubated with rabbit antibodies against C-3/9, CC-3/9, Cyto C and β-actin at 4 °C overnight, respectively, and then treated with anti-rabbit secondary antibody (1: 10,000) for 1 h at room temperature. Subsequently, the immunoreactive bands were visualized as mentioned above. β-actin was detected as a housekeeping protein control.

### In vivo anticancer effect in GBM stem cells xenografts

X01 orthotopic GBM stem cells bearing mice were randomly divided into five groups ($n = 9$, 5 for monitoring survival, 4 for histological analysis and western blotting) at day 10 post-implantation, followed by receiving an i.v. injection of HM-NPs@G, CM-NPs@G, MM-NPs@G, and free Gboxin (3 mg Gboxin equiv. kg$^{-1}$) or PBS every three days for totally five doses. After the injection, the mice were anesthetized and their tumor luminescence intensity was monitored by the Lumina IVIS III system. The relative photon flux was normalized to initial intensity, $I/I_0$ ($I_0$ is the bioluminescence intensity at day 10). On day 25, the treatment was terminated and four mice from each group were sacrificed for histological analysis and western blot analysis. The body weight of mice was individually measured every three days and the Kaplan-Meier survival curve was recorded during the period. The TUNEL and western blotting were evaluated similarly as above.

In order to compare the anti-GBM effect of HM-NPs@G and TMZ, we have performed the treatment which includes groups of PBS, free TMZ by oral administration, and HM-NPs@G on the orthotopic stem cell X01 models with the final dosage of Gboxin was 5 mg kg$^{-1}$ ($n = 8$, 5 for monitoring survival, 3 for histological analysis). After the injection, the mice were anesthetized and their tumor luminescence intensity was monitored by the Lumina IVIS III system. The relative photon flux was normalized to initial intensity, $I/I_0$ ($I_0$ is the bioluminescence intensity at day 7). On day 22, the treatment was terminated and 3 mice was sacrificed for hematoxylin and eosin staining and TUNEL analysis. The body weight of mice was individually measured every three days and the Kaplan-Meier survival curve was recorded during the period.

### Blood biochemistry analysis

Healthy Balb/c mice (female, 6–8 weeks) were randomly divided into three treatment groups ($n = 3$). HM-NPs@G, free Gboxin (5 mg Gboxin equiv. kg$^{-1}$) and PBS were i.v. injected into mice via the tail vein. Blood was collected via eye socket bleeding at prescribed time points post injection. For blood biochemistry examination, whole blood was centrifuged at 800 g for 5 min to collect serum for analysis. Standard blood chemistry parameters were analyzed using a kit from Wuhan Servicebio Technology Co. Ltd. on an automated chemistry analyzer (Chemray 240 Rayto lnc.). Blood cell parameters were analyzed with an automated blood cell analyzer (BC-2800Vet-Mindray Inc.). Meanwhile, at prescribed time points after injection, animals were anesthetized. The livers and kidneys were collected and these tissues were homogenized in 1 mL of ice-cold TRIzol reagent (Invitrogen) according to the protocol of the manufacturer. Reverse transcription and qPCR were carried out by following reverse transcription protocol (Takara) and SYBR Green Gene Expression Assays Protocol (Takara) with the Roche LightCycler 480 RT-PCR System. GAPDH was used as an endogenous housekeeping gene to normalize the target mRNA. The mRNA expression level was calculated based on comparative Ct method ($2 - \Delta\Delta Ct$).

### Statistics and reproducibility

All data are given as mean ± SD. Results were analyzed by using Microsoft Excel (2016), Origin 2021 and GraphPad Prism software 8.0. Differences between two groups were assessed using the Student's $t$ tests. For multiple comparisons, statistical significance was analyzed using one-way analysis of variance (ANOVA).

### Reporting summary

Further information on research design is available in the Nature Portfolio Reporting Summary linked to this article.

## Data availability

All data generated or analyzed during this study are available within the Article, its Supplementary Information file and the Source data file. Source data are provided with this paper.

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

## Acknowledgements

This work was supported by National Natural Science Foundation of China (NSFC U2004171, 32271463 and 32071388), the National Key Technologies R&D program of China (2018YFA0209800), Program for Science & Technology Innovation Talents in Universities of Henan Province (21HASTIT033), NHMRC Investigator Grants, Program of Technology Innovation Team in Colleges and Universities of Henan Province (21IRTSTHN028). Figure 2h, Fig. 4a, d, h, Fig. 5a, Fig. 6a and Supplementary Fig. 20a were created in Biorender.com. The authors acknowledge the Beijing Super Cloud Computing Center (BSCC) for its high performance and AI computing resources for Fig. 1e.

## Author contributions

Y.Z. and Y.S. contributed equally to this study. Y.Z. and B.S. designed the experiments, supervised the project, and revised the manuscript. Y.S. performed in vitro and in vivo experiments and prepared the manuscript. Y.W. and D.Z. helped to establish the orthotopic brain tumour models. H.Y. and X.W. helped to calculate the interaction energy of polymers and the drug. M.Z. commented and reviewed the manuscript. All authors participated in discussions throughout the project.

## Competing interests

The authors declare no competing interests.
