## [Peer Review File · Nature Communications]

Cancer cell-mitochondria hybrid membrane coated Gboxin-loaded nanomedicines for orthotopic glioblastoma inhibitionREVIEWER COMMENTS

Reviewer #1 (Remarks to the Author): with expertise in glioblastoma, metabolism

The authors prepared a "hybrid cell membrane camouflaged biomimetic" Gboxin in order to enhance brain tumor penetration. Gboxin was introduced as an inhibitor of complex V of the respiratory chain (oxidative phosphorylation) by another group a couple of years ago. The authors studied the in vitro efficacy, performed pharmacokinetics and assessed in vivo efficacy, using two orthotopic models of GBMs. There are some improvements necessary:

1. The U87 GBM model is highly discouraged to be used since its origin has been questioned.
2. The results in the orthotopic stem cell GBM model (Figure 6) is modest. They should test whether the efficacy is enhanced by the standard of care (radiation and temozolomide) (in vivo). This is a critical point.
3. They need to prove that actually Gboxin is really on-target in vitro and in vivo and its anti-glioma effects are not produced by some off-target effects.
4. Figure 2J is hard to understand. Why does the total level of both caspase 9 and caspase 3 rise with the treatment? It is understood though that the cleavage products should increase following treatment with Gboxin and its different formulations.

Reviewer #2 (Remarks to the Author): with expertise in glioblastoma, metabolism

in this manuscript, Yan zou et al present an interesting study shows that a Gboxin loaded nanoparticle camouflaged with a cell-mitochondria hybrid membrane (HM-NPs@G) not only improved the pharmacokinetic profile of Gboxin, but also provided the BBB permeability and enhanced cancer and mitochondria targeting of this compound. However, several major issues need to be clarified:

- 1- In the hybrid membrane, can the cancer cell membrane be replaced by the membrane

from other cancer cells or even wildtype cells?

2- Is this cell-mitochondria hybrid membrane a monolayer membrane? What's the percentage of mitochondrial inner membrane and outer membrane in this hybrid membrane?

3- Why cancer cell membrane can specifically inhibit ZO-1 and claudin-5 in BBB?

Minor:

1- More samples should be provided in Fig. s10.

2- Fig. 4i need quantification.

Reviewer #3 (Remarks to the Author): with expertise in nanoparticles, glioblastoma

In this manuscript, the authors developed a hybrid cell membrane camouflaged biomimetic Gboxin nanomedicine (HM-NPs@G) by coating cancer cell-mitochondria hybrid membrane (HM) on the surface of Gboxin loaded nanoparticles. The hybrid membrane camouflaging endowed HM-NPs@G with some features including improved pharmacokinetic profile, efficient BBB permeability, homotypic dual tumor cell and mitochondria targeting, and effective mitochondria damage. Despite of the adequate in vivo evaluation for the antitumor therapeutic efficacy, some points require further clarification. Therefore, I sincerely recommend major reversion. More detailed comments are provided below.

1. Förster resonance energy transfer (FRET) was utilized to demonstrate that cancer cell-mitochondria hybrid membrane was successfully camouflaged on ROS-responsive nanoparticles. As shown in Figure 1c, hybrid membrane camouflaged on NPs could not be clearly observed. The authors should use SIM or STED to demonstrate this. Therefore, authors should select other dyes to anchor on the cell membrane instead of load dyes into hydrophobic part of cell membrane, which is benefiting for accurately observing the phenomenon of membrane fusion. More importantly, upon laser irradiation, the emission light of DiO would excite the DiL owing to spectral overlap between DiO and DiL, thus resulting in the decreased fluorescence intensity of DiO. Why could authors observe bright fluorescence in the DiO fluorescence channel? The authors should provide the spectrums of

various cell membranes (cancer cell membrane, mitochondria membrane, and hybrid membrane) having been labeled with dyes.

2. In Figure 2i, the typical mitochondrial morphology could not be observed in the PBS group. Unhealthy glioblastoma cells cannot be used as control group.

3. Gboxin rapidly and irreversibly compromises oxygen consumption in glioblastoma cells (Nature, 2019, 567(7748), 1-6). Thus, authors should perform the experiment about oxygen consumption for further confirming the specific inhibition to mitochondria.

4. It's mentioned that the hybrid membrane camouflaging endowed HM-NPs@G with homotypic mitochondria targeting, thus authors should confirm that the loaded Gboxin could be precisely delivered to the mitochondria. Fluorescent labeling of Gboxin can refer to the methods in the reference (Nature, 2019, 567(7748), 1-6).

5. Why did authors use DiR in the experiment of biodistribution, whereas use Cy5 in ex vivo imaging of main organs? As shown in Figure 3c, HM-NPs exhibited obviously strong fluorescence in kidney, which was similar to the kidney in free Cy 5 group. It might be attributed to the leakage of Cy5 from HM-NPs according to the release results in Figure 1h. Moreover, there is controversial about using lipophilic dye (DiR) to label cell membrane-based NPs because the dye will label all the bilayer membranes. The dye may not be digested by the cells and has a long half-life in vivo, leading to artifact effects. Thus, the authors should select other method, such as anchoring dye on the cell membrane, to label the HM-NPs for investigating biodistribution in vivo.

6. Hydrophobic species could be loaded into hydrophobic part of hybrid cell membrane according to previous studies. Why did authors use the polymer nanoparticles for loading hydrophobic Gboxin? Moreover, was this polymer nanoparticles biodegradable? How would it be cleared in vivo?

7. It's mentioned that GBM cell membrane decorated NPs traverse the BBB by modulating the tight junctions between endothelial cells. What key components on the hybrid cell membrane opened the tight connection between endothelial cells?

8. U87MG is not an ideal model system of glioblastoma. It is known that these cell lines (e.g. U87MG, U251, GL261, and et al.) used to establish glioblastoma model do not represent the actual disease very well. Therefore, experiments, using such cell lines, are discouraged. Authors should discuss this point at least.

9. In Figure 5h, the image of TUNEL is not representative in HM-NPs@G group. Authors

should provide the large view images in all groups.

Reviewer #4 (Remarks to the Author): with expertise in nanoparticles, glioblastoma

An interesting example of anti-glioma treatment is proposed, based on cancer cells /mitochondria membrane camouflage of ROS-responsive polymeric nanoparticles delivering Gboxin, an inhibitor of oxidative phosphorylation. Some points that need improvements are listed in the following:

- Cell membrane and mitochondrial membrane characterization is not adequate: Western blotting or mass spectroscopy (see, for example, [10.1016/j.matdes.2020.108742](https://doi.org/10.1016/j.matdes.2020.108742)) on major proteins involved in targeting is needed (just Bcl2 is provided).
- While confocal analysis supports mitochondrial co-localization, electron microscopy (TEM) is warmly suggested to confirm this hypothesis.
- Figure 2i: what are the red arrows indicating?
- In vitro BBB model is oversimplified: astrocytes should be at least co-cultured with endothelial cells.
- ATP levels in cells should be evaluated upon the different treatments.
- Characterization (in terms of stemness markers) of glioma stem cells is missing.

Point-by-Point Responses to Reviewers' Comments

We would like to thank the reviewers for the constructive comments. We have carefully revised our manuscript according to the reviewers' comments. The following is a list of changes we have made.

Reviewer #1

The authors prepared a "hybrid cell membrane camouflaged biomimetic" Gboxin in order to enhance brain tumour penetration. Gboxin was introduced as an inhibitor of complex V of the respiratory chain (oxidative phosphorylation) by another group a couple of years ago. The authors studied the in vitro efficacy, performed pharmacokinetics and assessed in vivo efficacy, using two orthotopic models of GBMs. There are some improvements necessary:

1. The U87 GBM model is highly discouraged to be used since its origin has been questioned.

Response: We thank the reviewer for this constructive comment. We agree with the reviewer that U87MG is not the perfect model to mimic the pathologies of GBM patients. In this study, we just choose U87MG as a basic GBM model to initially evaluate our hybrid membrane coated nanomedicine HM-NPs@G *in vitro* and *in vivo* as the proof-of-concept which is well-adopted in other pre-clinic studies (ACS Nano, 2023,17:240-250.; Sci Adv, 2022, 18: eabl4923.; Nat Commun, 2020, 11:295.; Nat Commun, 2019, 10: 2448.; Angew Chem Int Ed Engl, 2022, 61: e202214786.). To further systematically assess the treatment outcome of HM-NPs@G, we did employ the patient-derived GBM stem cells (X01, GSCs) model (**Figure 6**).

To clarify, we have added the relevant discussion in **Page 16 Lines 18-25**: "Though U87MG is not the perfect model to mimic the pathologies of GBM patients, there is no doubt to employ it as a GBM model in the initial proof-of-concept studies to verify the anti-tumour effect in preclinical studies (ACS Nano, 2023,17:240-250.; Sci Adv, 2022, 18: eabl4923.; Nat Commun, 2020, 11:295.; Nat Commun, 2019, 10: 2448.; Angew Chem Int Ed, 2022, 61: e202214786.). Therefore, we firstly demonstrated the efficacy of HM-NPs@G towards the U87MG model, and the results strongly suggested that HM-NPs@G could inhibit tumour growth by activating mitochondria-related apoptosis. Furthermore, we adopted patient-derived GBM stem cells (GSCs) to establish mouse models and the results showed that the therapeutic outcomes are in line with that of U87MG model, further confirmed the superior effects of this biomimetic nanomedicine which is a potential formulation be translated in clinical for treating GBM."

2. *The results in the orthotopic stem cell GBM model (Figure 6) is modest. They should test whether the efficacy is enhanced by the standard of care (radiation and temozolomide) (in vivo). This is a critical point.*

Response: As suggested by the reviewer, we have performed the treatment of HM-NPs@G on the orthotopic stem cell GBM models by adopting TMZ (the first-line GBM drug in clinic) as a control. The results showed that HM-NPs@G nanomedicines showed enhanced anti-tumour efficacy and extended median survival lifespan (65 d *versus* 40 d) as compared with the free TMZ (**Figure 7**), further demonstrating the excellent GBM inhibition of HM-NPs@G. Accordingly, we have added the details in **Page 14 Lines 19-26 & Page 15 Lines 1-7**: In order to compare the anti-GBM effect of HM-NPs@G and GBM first-line drug TMZ, we have performed the treatment of HM-NPs@G towards orthotopic GSC models by using the TMZ as a control via oral administration to mimic the clinical treatment (**Figure 7a**). The results showed that HM-NPs@G had a significantly enhanced anti-tumour effect compared with free TMZ. Interestingly, though the tumours could be restrained to some extent after being treated with free TMZ, the tumours erupted promptly when the injection was terminated. In sharp contrast, HM-NPs@G group exhibited continuous tumour inhibition in a longer period (**Figure 7b, c**). The body weight of mice had a slight increase for HM-NPs@G while a subsequent and sharp reduction for TMZ and PBS treatments, supporting the effective anti-GBM effects of HM-NPs@G (**Figure 7d**). Importantly, the median survival time of mice treated with HM-NPs@G (65 d) was significantly longer in comparison to that of free TMZ and PBS group, which were 40 d and 19 d, respectively (**Figure 7e**). The images of tumour-bearing brain slices stained with H&E showed that the tumour volume of HM-NPs@G group was the smallest (**Figure 7f**). The TUNEL results indicated that more apoptotic GBM tumour cells were observed in the slices followed HM-NPs@G treatment (**Figure 7g**). Taken together, the HM-NPs@G nanomedicines demonstrate enhanced therapeutic outcome than free TMZ, providing a potential alternative drug to be used in clinic.

Figure 7. *In vivo* antitumour activity of HM-NPs@G in mice bearing GBM stem cells (GSCs) xenografts. **a**, Schematic illustration of the establishment of the PDX-derived GBM GSCs orthotopic model. **b**, Time course luminescence images of mice bearing orthotopic X01-

Luc tumours following treatment with HM-NPs@G, free TMZ or PBS (n=5). The mice were intravenously injected with HM-NPs@G or orally injected with TMZ (the dosage for both the Gboxin and TMZ were 5 mg equiv. kg⁻¹) on day 7, 10, 13, 16 and 19 post tumour implantation. **c**, Quantified luminescence levels of mice using the Lumina IVIS III system (n = 5). **d**, Body weight profile (n = 5 biologically independent samples). Data are presented as mean ± SD (one-way ANOVA and Tukey multiple comparisons tests). **e**, Mice survival rate curves. Statistical analysis: HM-NPs@G vs. TMZ, ***p* < 0.01, HM-NPs@G vs. PBS, ****p* < 0.001. (n=5, Kaplan-Meier analysis, log-rank test). **f**, H&E images of the whole brain excised from mice treated as described above on day 22. **g**, Histological analysis using TUNEL assay. Green: apoptotic cells; blue: Hoechst-stained cell nuclei (Scale bar = 100 μm).

3. *They need to prove that actually Gboxin is really on-target in vitro and in vivo and its anti-glioma effects are not produced by some off-target effects.*

Response: We appreciate the reviewer's constructive comment. To evaluate if the excellent anti-tumour effects of HM-NPs@G result from the homologous targeting capability of hybrid membrane coating, we extracted the mitochondria of U87MG cells treated with different nanomedicines and quantified the Gboxin in the mitochondria. The results showed that approximately 23.4 μg/mL Gboxin was accumulated in the mitochondria of HM-NPs@G treated cells, which was significantly higher than that of single membrane coated nanomedicines CM-NPs@G (16.6 μg/mL) and MM-NPs@G (12.1 μg/mL), while Gboxin could not be detected for the free Gboxin and NPs@G treatments (**Figure S15**). These all indicate the excellent homotypic targeting of hybrid membrane coated nanomedicines, the on-target effects directly benefit to the tumour cell growth inhibition.

Correspondingly, we have included these results in **Page 9 Lines 10-16: On-target effects of HM-NPs@G assay in vitro**. The on-target effects of HM-NPs@G were evaluated by isolating and quantifying the Gboxin in mitochondria determined by HPLC. The results showed that about 23.4 μg/mL Gboxin was accumulated in the mitochondria of HM-NPs@G treated cells, which was significantly higher than that of single membrane coated nanomedicines CM-NPs@G (16.6 μg/mL) and MM-NPs@G (12.1 μg/mL), while Gboxin could not be detected for the free Gboxin and NPs@G treatments (**Figure S15**). These all indicate the excellent homotypic targeting of hybrid membrane coated nanomedicines and the on-target effects directly benefit to the tumour cell growth inhibition.

Figure S15. Gboxin accumulation amount analysis in the mitochondria of U87MG cells treated with different formulations *in vitro*. The accumulation amount of Gboxin was detected by HPLC in mitochondria of U87MG cells treated with HM-NPs@G, MM-NPs@G, CM-NPs@G, NPs@G and free Gboxin for 24 h (n=3). Data are presented as mean \pm SD (one-way ANOVA and Tukey multiple comparisons tests).

The targeting capability of hybrid membrane is further evaluated by encapsulating the inorganic upconversion nanoparticles (UCNPs) and observed with bio-TEM (transmission electron microscopy). The images showed that the majority of HM-UCNPs were accumulated in mitochondria, while few signals were observed in the cytoplasm or other subcellular organelles (**Figure S6**), further demonstrating the superb mitochondria targeting ability of hybrid membrane coating strategy. In accordance, the relevant description has been added in **Page 6 Lines 22-23 & Page 7 Lines 1-7**: Furthermore, the targeting ability of HM was further assessed by treating cells with HM encapsulating upconversion nanoparticles (HM-UCNPs) and be observed with Bio-TEM. The results showed that notable UCNPs were delivered into U87MG cells by HM-UCNPs and CM-UCNPs with active targeting of CM, while much fewer UCNPs were observed in MM-UCNPs and negligible UCNPs were observed in the naked UCNPs treating cells (**Figure S6**), indicating the homologous targeting capability of CM. Meanwhile, single CM modified UCNPs showed the limited capability to target and accumulate in the mitochondria, evidenced by abundant UCNPs in the cytoplasm rather than the mitochondria. Remarkably, the majority of HM-UCNPs were located in the mitochondria, suggesting the dual-targeting of HM to both tumour cells and mitochondria organelles.

Collectively, these results support that the good homologous targeting capability of hybrid membrane could directly lead to the enhanced anti-tumour effects.

Figure S6. Co-localization analysis by Bio-TEM. TEM images of mitochondria in U87MG cells treated with HM-UCNPs, CM-UCNPs, MM-UCNPs and bare UCNPs. UCNPs: 1 mg/mL. Scale bar = 500 nm. The red arrows indicated the UCNPs.

4. *Figure 2J is hard to understand. Why does the total level of both caspase 9 and caspase 3 rise with the treatment? It is understood though that the cleavage products should increase following treatment with Gboxin and its different formulations.*

Response: We would like to thank the reviewer for the constructive comment. To ensure the reliability of the regulation of apoptosis-related proteins, we repeated the western blotting in cells treated with nanomedicines for more than 3 times and obtained the similar trend as displayed in **Figure R1**. That's mainly due to the fact that the membrane coated nanomedicines possess higher mitochondria targeting and accumulating capability that lead to severe structure damage of mitochondria (**Figure 2i**), which may stimulate the expression of apoptosis relevant proteins (both total and cleaved types). The other reports also found the total caspase proteins are up-regulated followed the treatments (Small, 2018, 15: 1803428.; Int J Biol Macromol, 2019, 130: 997-1008.; ACS Appl Mater Interfaces, 2022, 14:42541-42557.; ACS Nano, 2019, 13:2501-2510.). For instance, the levels of caspase-8 and caspase3/7 of MDA-MB-231 TNBC cells treated with PVX-HisTRAIL achieved 2-fold and 2.9-fold enhancement than the PBS controls (ACS Nano, 2019, 13:2501-2510.).

Figure R1. Western blotting analysis of apoptosis-related proteins in U87MG cells treated with free Gboxin, NPs@G, CM-NPs@G, MM-NPs@G, or HM-NPs@G) for 72 h.

Reviewer #2 (Remarks to the Author): with expertise in glioblastoma, metabolism

In this manuscript, Yan zou et al present an interesting study shows that a Gboxin loaded nanoparticle camouflaged with a cell-mitochondria hybrid membrane (HM-NPs@G) not only improved the pharmacokinetic profile of Gboxin, but also provided the BBB permeability and enhanced cancer and mitochondria targeting of this compound. However, several major issues need to be clarified:

1- *In the hybrid membrane, can the cancer cell membrane be replaced by the membrane from other cancer cells or even wildtype cells?*

Response: We would like to thank the reviewer's nice comment. Yes, the cancer membrane can be replaced by any other membranes, including red blood membranes, other types of cancer cell membranes, immune cell membranes and so on.

2- *Is this cell-mitochondria hybrid membrane a monolayer membrane? What's the percentage of mitochondrial inner membrane and outer membrane in this hybrid membrane?*

Response: The hybrid membrane is a single-membrane lipid bilayer that can be observed clearly by TEM images and determined by the increased size of HM-NPs@G as compared with non-membrane control. Moreover, it has been reported that biomimetic membrane camouflaging would increase by 10-20 nm, with around 8 nm thickness of bilayer lipid (Small, 2021, 17: 2006484). In terms of the ratio between inner and outer membrane of mitochondria membrane, it may be the same as the natural mitochondria because we isolated the total mitochondrial membranes containing both inner and outer membranes.

In accordance, we have added a couple of more sentences in **Page 4 Lines 8-10 & Page 4 Line 10-12**: In terms of the ratio of MM between inner and outer membrane, it may be the same as the natural mitochondria because we isolated the total mitochondrial membranes containing both inner and outer membranes. The core-shell structure of the developed HM-NPs@G was confirmed with the transmission electron microscopy (TEM) (**Figure 1b**), indicating the hybrid membrane is a single-membrane lipid bilayer which is agree with the reported results (Small, 2021, 17: 2006484.; Nat Commun, 2022, 13: 4214.).

3- *Why cancer cell membrane can specifically inhibit zo-1 and claudin-5 in BBB?*

Response: We appreciate the reviewer's constructive comment. There are multiple interaction molecules on the surface of cancer membrane including integrin, Mac-1 and other special proteins, which facilitate the membrane coated nanoparticles to traverse the BBB by modulating the tight junctions (Biomaterials, 2019, 211: 48-56.; Nanoscale, 2020, 12: 15473.; Nat Commun, 2022, 13: 4212.). As reported, these special membrane proteins can target the intercellular adhesion molecule-1 (ICAM-1) receptors on endothelial cells, indirectly enhancing the BBB permeability via decreasing the expression of tight junction proteins including ZO-1 and Claudin-5.

Accordingly, we have added one more sentence in **Page 11 Lines 14-16**: There are multiple interaction molecules on the surface of cancer membrane including integrin, Mac-1 and other special proteins, which facilitate the membrane coated nanoparticles to traverse the BBB by modulating the tight junctions (Biomaterials, 2019, 211: 48-56.; Nanoscale, 2020, 12: 15473.; Nat Commun, 2022, 13: 4212.).

Minor:

1- *More samples should be provided in Fig. s10.*

2- *Fig. 4i need quantification.*

Response: We are appreciated and agree with the reviewer's nice comments. As suggested, we have added three samples in **Figure S16** (original Figure S10).

Additionally, the expression levels of tight junction-related proteins have been quantified in **Figure 4i, 4j**.

Figure S16. Ex vivo imaging analysis. Cy5 fluorescence images of GBM brains taken from U87MG-Luc bearing mice at 6 h post *i.v.* injection of HM-NP@Cy5 (1 mg Cy5 equiv. kg⁻¹).

Figure 4i, j. Western blot images of ZO-1 and claudin-5 expression and their qualified levels in hCMEC/D3 cells during-treatment (i) and post-treatment (j) with HM-NPs, MM-NPs, CM-NPs, bare NPs or PBS.

Reviewer #3 (Remarks to the Author): with expertise in nanoparticles, glioblastoma

In this manuscript, the authors developed a hybrid cell membrane camouflaged biomimetic Gboxin nanomedicine (HM-NPs@G) by coating cancer cell-mitochondria hybrid membrane (HM) on the surface of Gboxin loaded nanoparticles. The hybrid membrane camouflaging endowed HM-NPs@G with some features including improved pharmacokinetic profile, efficient BBB permeability, homotypic dual tumour cell and mitochondria targeting, and effective mitochondria damage. Despite of the adequate in vivo evaluation for the antitumour

therapeutic efficacy, some points require further clarification. Therefore, I sincerely recommend major reversion. More detailed comments are provided below.

1. Förster resonance energy transfer (FRET) was utilized to demonstrate that cancer cell-mitochondria hybrid membrane was successfully camouflaged on ROS-responsive nanoparticles. As shown in Figure 1c, hybrid membrane camouflaged on NPs could not be clearly observed. The authors should use SIM or STED to demonstrate this. Therefore, authors should select other dyes to anchor on the cell membrane instead of load dyes into hydrophobic part of cell membrane, which is benefiting for accurately observing the phenomenon of membrane fusion. More importantly, upon laser irradiation, the emission light of DiO would excite the DiL owing to spectral overlap between DiO and DiL, thus resulting in the decreased fluorescence intensity of DiO. Why could authors observe bright fluorescence in the DiO fluorescence channel? The authors should provide the spectrums of various cell membranes (cancer cell membrane, mitochondria membrane, and hybrid membrane) having been labeled with dyes.

Response: We are really grateful for the kind comments. Firstly, we would like to clarify that the result of **Figure 1c** aims to demonstrate the CM and MM were successfully fused by adopting the DiO and DiL as reported (Nature, 2022, 612: 546-554). We agree with the reviewer that the spectrum overlap between DiO (484~501 nm) and DiL (551~569 nm) may affect the fluorescence intensity. To avoid the interference, we chosen DiD (644~665 nm) instead of DiL, to stain the membrane. The confocal images showed that HM displayed yellow due to the co-localization of MM (green) and CM (red) (**Figure 1c**), suggesting the successful preparation of HM. Accordingly, we have modified the description in **Page 4 Lines 13-14**: **The successful fusion of CM and MM was demonstrated by confocal microscopy as co-localization of specific CM (red) and MM (green) fluorescent signals were observed (Figure 1c).**

Figure 1c, CLSM images of fabricated hybrid membrane (HM) vesicles. Mitochondrial membranes (MM) were labeled with DiO (green) and cancer cell membranes (CM) were labeled with DiD (red). The merged image showed yellow fluorescence with similar

morphology to MM and CM confirming the successful fabrication of HM vesicles. Scale bar = 20 μm .

2. In Figure 2i, the typical mitochondrial morphology could not be observed in the PBS group. Unhealthy glioblastoma cells cannot be used as control group.

Response: We would like to thank the reviewer for this constructive comment. As suggested, we have further observed the construction of U87MG cells treated with HM-NPs@G by using the microglia normal cell as a control. The results showed that the normal cells have typical mitochondria morphology, while mitochondria constructions of U87MG cells were damaged obviously after being treated with HM-NPs@G (**Figure S14**). In accordance, we have added one more sentence in **Page 8 Line 19-22**: To further identify the structure of mitochondria, we used the microglia cell as a control to demonstrate the typical mitochondria morphology that is significantly different from damaged mitochondria in U87MG cells induced by the HM-NPs@G (**Figure S14**), indicating the therapeutic effect of HM-NPs@G.

Figure S14. Mitochondria morphology analysis of normal cells and HM-NPs@G treated GBM cells using Bio-TEM. TEM images of mitochondria in normal cells (HA1800) and U87MG cells treated with HM-NPs@G. The red arrows indicated the damaged construction of mitochondria after treatment of nanomedicines. Scale bar = 500 nm.

3. Gboxin rapidly and irreversibly compromises oxygen consumption in glioblastoma cells (*Nature*, 2019, 567(7748), 1-6). Thus, authors should perform the experiment about oxygen consumption for further confirming the specific inhibition to mitochondria.

Response: As suggested by the reviewer, we have studied the oxygen consumption ratio (OCR) in U87MG cells and glioblastoma stem cells (GSCs, X01) treated with different nanoparticles.

The results showed that HM-NPs@G led to the OCR of 18.87% and 10.44% for U87MG and X01 cells, which were the lowest as compared with other groups (**Figure S13**), suggesting that superior anti-tumour effects of HM-NPs@G benefit from the consumption of oxygen. Intriguingly, much lower OCR was detected in X01 than U87MG cells, indicating the sensitivity of GSCs cells to Gboxin and in accordance with the lower IC₅₀ concentration. Correspondingly, the results were included in **Page 8 Lines 9-16**: In addition, the oxygen consumption rate (OCR) was conducted in U87MG and X01 cells. Both U87MG cells and X01 cells were subjected to different formulations include free Gboxin, NPs@G, CM-NPs@G, MM-NPs@G, and HM-NPs@G. The results showed that HM-NPs@G led to the OCR of 18.87% and 10.44% for U87MG and X01 cells, which were the lowest as compared with other groups (**Figure S13**), suggesting that superior anti-tumour effects of HM-NPs@G benefit from the consumption of oxygen. Intriguingly, much lower OCR was detected in X01 than U87MG cells, indicating the sensitivity of GSCs cells to Gboxin and in accordance with the lower IC₅₀ concentration.

Figure S13. Oxygen consumption rate analysis in U87MG and GSCs X01 cells *in vitro*. The rate of oxygen consumption within 2 h in U87MG (a) and X01 cells (b) treated with HM-NPs@G, MM-NPs@G, CM-NPs@G, NPs@G and free Gboxin was analysed. Data are presented as mean \pm SD (one-way ANOVA and Tukey multiple comparisons tests).

4. It's mentioned that the hybrid membrane camouflaging endowed HM-NPs@G with homotypic mitochondria targeting, thus authors should confirm that the loaded Gboxin could be precisely delivered to the mitochondria. Fluorescent labeling of Gboxin can refer to the methods in the reference (Nature, 2019, 567(7748), 1-6).

Response: We appreciate the reviewer's nice comment. To evaluate if the loaded Gboxin could be delivered precisely to the mitochondria, we extracted the mitochondria of U87MG cells treated with different nanomedicines and quantified the Gboxin in the mitochondria. The results showed that about 23.4 $\mu\text{g}/\text{mL}$ Gboxin was accumulated in the mitochondria of HM-NPs@G treated cells, which is significantly higher than that of single membrane coated nanomedicines CM-NPs@G (16.6 $\mu\text{g}/\text{mL}$) and MM-NPs@G (12.1 $\mu\text{g}/\text{mL}$), while Gboxin could not be detected for the free Gboxin and NPs@G treatments (**Figure S15**). These all indicate the excellent homotypic targeting of hybrid membrane coated nanomedicines, the on-target effects directly benefit the tumour cell growth inhibition.

Correspondingly, we have included these results in **Page 9 Lines 10-16: On-target effects of HM-NPs@G assay *in vitro***. The on-target effects of HM-NPs@G were evaluated by isolating and quantifying the Gboxin in mitochondria determined by HPLC. The results showed that about 23.4 $\mu\text{g}/\text{mL}$ Gboxin was accumulated in the mitochondria of HM-NPs@G treated cells, which was significantly higher than that of single membrane coated nanomedicines CM-NPs@G (16.6 $\mu\text{g}/\text{mL}$) and MM-NPs@G (12.1 $\mu\text{g}/\text{mL}$), while Gboxin could not be detected for the free Gboxin and NPs@G treatments (**Figure S15**). These all indicate the excellent homotypic targeting of hybrid membrane coated nanomedicines and the on-target effects directly benefit to the tumour cell growth inhibition.

Figure S15. Gboxin accumulation ratio analysis in the mitochondria of U87MG cells treated with different formulations *in vitro*. The accumulation ratio of Gboxin was detected with HPLC in mitochondria of U87MG cells treated with HM-NPs@G, MM-NPs@G, CM-

NPs@G, NPs@G and free Gboxin for 24 h (n=3). Data are presented as mean \pm SD (one-way ANOVA and Tukey multiple comparisons tests).

The targeting capability of hybrid membrane is further evaluated by encapsulating the inorganic upconversion nanoparticles (UCNPs) with bio-TEM (transmission electron microscopy). The images showed that the majority of HM-UCNPs were accumulated in the mitochondria, while few signals were observed in the cytoplasm or other subcellular organelles, further demonstrating the superb mitochondria targeting ability of hybrid membrane coating strategy. In accordance, the relevant description has been added in **Page 6 Lines 22-23 & Page 7 Lines 1-7**: Furthermore, the targeting ability of HM was further assessed by treating cells with HM encapsulating upconversion nanoparticles (HM-UCNPs) and being observed with Bio-TEM. The results showed that notable UCNPs were delivered into U87MG cells by HM-UCNPs and CM-UCNPs with active targeting of CM, while much fewer UCNPs were observed in MM-UCNPs and negligible UCNPs were observed in the naked UCNPs treating cells (**Figure S6**), indicating the homologous targeting capability of CM. Meanwhile, single CM modified UCNPs showed the limited capability to target and accumulate in the mitochondria, evidenced by abundant UCNPs in the cytoplasm rather than the mitochondria. Remarkably, the majority of HM-UCNPs were located in the mitochondria, suggesting the dual-targeting of HM to both tumour cells and mitochondria organelles.

Figure S6. Co-localization analysis by Bio-TEM. TEM images of mitochondria in U87MG cells treated with HM-UCNPs, CM-UCNPs, MM-UCNPs and bare UCNPs. UCNPs: 1 mg/mL. Scale bar = 500 nm. The red arrows indicated the UCNPs.

Collectively, these results support that cancer cell and mitochondria hybrid membrane coating endows the nanoparticles dual-targeting capability that can precisely deliver drugs to tumour cells and mitochondria sites.

5. *Why did authors use DiR in the experiment of biodistribution, whereas use Cy5 in ex vivo imaging of main organs? As shown in Figure 3c, HM-NPs exhibited obviously strong fluorescence in kidney, which was similar to the kidney in free Cy 5 group. It might be attributed to the leakage of Cy5 from HM-NPs according to the release results in Figure 1h. Moreover, there is controversial about using lipophilic dye (DiR) to label cell membrane-based NPs because the dye will label all the bilayer membranes. The dye may not be digested by the cells and has a long half-life in vivo, leading to artifact effects. Thus, the authors should select other method, such as anchoring dye on the cell membrane, to label the HM-NPs for investigating biodistribution in vivo.*

Response: We are really appreciated the reviewer's constructive comment. We firstly would like to clarify that we used Cy5 for *ex vivo* imaging and staining of brain tumour slices as DiR cannot be detected by the CLSM (Zeiss 880) due to its long wavelength (excitation 754 nm, emission 778 nm).

Taking into account that the size of these nanomedicines is around 90 nm, they were excluded mainly through the kidney metabolism pathway (J Control Release, 2021, 10, 334: 127-137; Nat Commun, 2017, 12, 8: 878.) and that's why obvious Cy5 accumulation was observed in kidney.

To keep in line with the *ex vivo* imaging and avoid the confusion, we performed the *in vivo* imaging by loading Cy5 instead of DiR (**Figure 3b**), the results were consistent with DiR loaded nanoparticles that HM-NPs displayed the strongest fluorescence in the brain among all the groups, demonstrating the excellent BBB permeability and tumour targeting ability of HM-NPs. Moreover, obvious Cy5 was observed in mice treated with HM-NPs at 2 h and lasted for 24 h while pale fluorescence could be seen for free Cy5 at 6 h post injection due to the short blood circulation and fast elimination from the body. Therefore, we deduce the leakage of Cy5 may contribute to the Cy5 accumulation in the kidney, but the majority of Cy5 were encapsulated into nanoparticles with a longer blood half-life.

The images have been replaced in Figure 3b and the results are revised in **Page 10 Lines 5-10**:
To confirm whether cancer membrane camouflaging promoted specific NPs accumulation in GBM, near-infrared dye Cy5 loaded NPs were systemically injected into luciferase expressing U87MG (U87MG-Luc) tumour-bearing nude mice and monitored in real-time with Cy5

fluorescence. An obviously stronger red fluorescence was observed in the brain of mice treated with HM-NPs and CM-NPs 6 h post-injection which remained detectable up to 24 h (**Figure 3b**), indicating that both HM-NPs and CM-NPs were able to traverse the BBB to accumulate in GBM tissue.

Figure 3b, *In vivo* fluorescence images of orthotopic U87MG-Luc human GBM tumour bearing nude mice following a single tail vein injection of HM-NPs@Cy5 (2 mg Cy5 equiv. kg^{-1}).

6. *Hydrophobic species could be loaded into hydrophobic part of hybrid cell membrane according to previous studies. Why did authors use the polymer nanoparticles for loading hydrophobic Gboxin? Moreover, was this polymer nanoparticles biodegradable? How would it be cleared in vivo?*

Response: We thank the reviewer for pointing out this. We agree with the reviewer that hydrophobic species could be loaded into hydrophobic materials as reported. Actually, we tried to load the Gboxin into the hybrid membrane directly, while the drug loading efficiency was terribly low and even undetectable attributes to the special chemical structure of Gboxin. Furthermore, we synthesized a couple of polymers to encapsulate Gboxin and calculated the interaction energy between polymers and Gboxin. The calculation results showed the interaction energy of PEG-PHB and Gboxin was the largest, indicating it could load Gboxin efficiently. The experimental loading content further confirmed that PEG-PHB polymeric

nanoparticles displayed a high drug loading content of 16.1%. Therefore, we used PEG-PHB based nanoparticles to encapsulate the Gboxin for the *in vitro* and *in vivo* study.

Moreover, this polymer nanocarrier is ROS-responsive which benefits to controlled drug release in a high level of ROS in the mitochondria of glioblastoma cells. Importantly, PEG-PHB is biocompatible and biodegradable, which would be degraded into PEG, pinacol borate and p-hydroxy-methylphenol and eliminated from the body as triggered by the ROS.

In accordance, we have added the discussion in **Page 5 Lines 4-5**: **It should be noted that the PEG-PHB polymer could be degraded into PEG, pinacol borate and p-hydroxy-methylphenol and further eliminated from the body.**

7. *It's mentioned that GBM cell membrane decorated NPs traverse the BBB by modulating the tight junctions between endothelial cells. What key components on the hybrid cell membrane opened the tight connection between endothelial cells?*

Response: We are really grateful for the constructive comment. The higher BBB penetration of HM-NPs was mainly achieved by cancer cell membrane rather than mitochondria membrane. There are multiple interaction molecules on the surface of cancer membrane including integrin, Mac-1 and other special proteins, which facilitate the membrane coated nanoparticles to traverse the BBB by modulating the tight junctions (Biomaterials, 2019, 211: 48-56.; Nanoscale, 2020, 12: 15473.; Nat Commun, 2022, 13: 4212.). As reported, these special membrane proteins can target the intercellular adhesion molecule-1 (ICAM-1) receptors on endothelial cells, indirectly enhancing the BBB permeability via decreasing the expression of tight junction proteins including ZO-1 and Claudin-5.

Accordingly, we have added the discussions in **Page 11 Lines 14-16**: **There are multiple interaction molecules on the surface of cancer membrane including integrin, Mac-1 and other special proteins, which facilitate the membrane coated nanoparticles to traverse the BBB by modulating the tight junctions (Biomaterials, 2019, 211: 48-56.; Nanoscale, 2020, 12: 15473.; Nat Commun, 2022, 13: 4212.).**

8. *U87MG is not an ideal model system of glioblastoma. It is known that these cell lines (e.g. U87MG, U251, GL261, and et al.) used to establish glioblastoma model do not represent the actual disease very well. Therefore, experiments, using such cell lines, are discouraged. Authors should discuss this point at least.*

Response: We thank the reviewer for this constructive comment. We agree with the reviewer that U87MG is not the perfect model to mimic the pathologies of GBM patients. In this study,

we just choose U87MG as a basic GBM model to initially evaluate our hybrid membrane coated nanomedicine HM-NPs@G *in vitro* and *in vivo* as the proof-of-concept which is well-adopted in other pre-clinic studies (ACS Nano, 2023,17:240-250.; Sci Adv, 2022, 18: eabl4923.; Nat Commun, 2020, 11:295.; Nat Commun, 2019, 10: 2448.; Angew Chem Int Ed Engl, 2022, 61: e202214786.). To further systematically assess the treatment outcome of HM-NPs@G, we did employ the patient-derived GBM stem cells (X01, GSCs) model (**Figure 6**).

To clarify, we have added the relevant discussion in **Page 16 Lines 18-25**: “Though U87MG is not the perfect model to mimic the pathologies of GBM patients, there is no doubt to employ it as a GBM model in the initial proof-of-concept studies to verify the anti-tumour effect in preclinical studies (ACS Nano, 2023,17:240-250.; Sci Adv, 2022, 18: eabl4923.; Nat Commun, 2020, 11:295.; Nat Commun, 2019, 10: 2448.; Angew Chem Int Ed, 2022, 61: e202214786.). Therefore, we firstly demonstrated the efficacy of HM-NPs@G towards U87MG model, and the results strongly suggested that HM-NPs@G could inhibit tumour growth by activating mitochondria-related apoptosis. Furthermore, we adopted patient-derived GBM stem cells (GSCs) to establish mice models and the results showed that the therapeutic outcomes are in line with that of U87MG model, further confirmed the superior effects of this biomimetic nanomedicine which is a potential formulation be translated in clinical for treating GBM.”

9. In Figure 5h, the image of TUNEL is not representative in HM-NPs@G group. Authors should provide the large view images in all groups.

Response: As suggested by the reviewer, we have provided the TUNEL images with the larger view. It showed that HM-NPs@G induced abundant apoptosis tumour cells that in accordance with the therapeutic outcomes.

Figure 6h, Histological analysis using the TUNEL assay. Green: apoptotic cells; blue: Hoechst-stained cell nuclei (Scale bar = 100 μ m).

Reviewer #4 (Remarks to the Author): with expertise in nanoparticles, glioblastoma

An interesting example of anti-glioma treatment is proposed, based on cancer cells /mitochondria membrane camouflage of ROS-responsive polymeric nanoparticles delivering

Gboxin, an inhibitor of oxidative phosphorylation. Some points that need improvements are listed in the following:

1. *Cell membrane and mitochondrial membrane characterization is not adequate: Western blotting or mass spectroscopy (see, for example, 10.1016/j.matdes.2020.108742) on major proteins involved in targeting is needed (just Bcl2 is provided).*

Response: As suggested by the reviewer, we have characterized the proteins on the cancer cell membrane and mitochondria membrane by the western blotting (Nano Lett, 2020, 20: 7; Trends Cell Biol, 2021, 31: 62-74; Proc Natl Acad Sci, 2017, 114: E9863-E9872). The results showed that the key proteins (Atlastin-1, EHD2 and Mito-fusion) related to mitochondria targeting and penetration were observed on mitochondria membrane (MM) and hybrid membrane coated NPs (HM-NPs). In addition, the proteins (EpCAM and Integrin αv) which play vital roles in cancer homologous targeting were observed on U87MG cancer cell membrane (CM). Furthermore, glioblastoma stem cell (GSCs, X01) membrane CM (X01) had CD44, one of stem markers, as well as EpCAM, both of which were helpful to target homotypic cells. Surprisingly, CD44 and Integrin αv also expressed on MM, which endows the MM-NPs targeting capability to GBM cells (**Figure 1d**). Moreover, Atlastin-1, which is closely related to the bio-membrane fusion, was observed on CM, MM and HM-NPs, facilitating the permeability to tumour cell and mitochondria. Accordingly, we have included these results in **Page 4 Lines 14-23 & Page 5 Line 1**: Afterwards, we have further characterized proteins on the CM and MM by the western blots. As shown in **Figure 1d**, the key proteins (Atlastin-1, EHD2 and Mito-fusion) related to mitochondria targeting and penetration were observed on mitochondria membrane (MM) and hybrid membrane coated NPs (HM-NPs). In addition, the proteins (EpCAM and Integrin αv) which play vital roles in cancer homologous targeting were observed on U87MG cancer cell membrane (CM). Furthermore, glioblastoma stem cell (GSCs, X01) membrane CM (X01) had CD44, one of stem markers, as well as EpCAM, both of which were helpful to target homotypic cells. Surprisingly, CD44 and Integrin αv also expressed on MM, which endow the MM-NPs targeting capability to GBM cells to some extent (**Figure 1d**). Moreover, Atlastin-1, which is closely related to the bio-membrane fusion, was expressed on CM, MM and HM-NPs, facilitating the permeability to tumour cell and mitochondria.

Figure 1d, Western blotting analysis cancer membrane and mitochondria membrane special targeting related proteins. i: HM-NPs (X01), ii: HM-NPs (U87MG), iii: NPs, iv: MM, v: CM (X01), vi: CM (U87MG).

2. While confocal analysis supports mitochondrial co-localization, electron microscopy (TEM) is warmly suggested to confirm this hypothesis.

Response: We appreciate the reviewer's nice comment. As suggested, the targeting capability of hybrid membrane is further evaluated by encapsulating the inorganic upconversion nanoparticles (UCNPs) and observed with bio-TEM (transmission electron microscopy). The images showed that the majority of HM-UCNPs were accumulated in the mitochondria, while few signals were observed in the cytoplasm or other subcellular organelles (**Figure S6**), further demonstrating the superb mitochondria targeting ability of hybrid membrane coating strategy. In accordance, the relevant description has been added in **Page 6 Lines 22-23 & Page 7 Lines 1-7**: Furthermore, the targeting ability of HM was further assessed by treating cells with HM encapsulating upconversion nanoparticles (HM-UCNPs) and be observed with Bio-TEM. The results showed that notable UCNPs were delivered into U87MG cells by HM-UCNPs and CM-UCNPs with active targeting of CM, while much fewer UCNPs were observed in MM-UCNPs and negligible UCNPs were observed in the naked UCNPs treating cells (**Figure S6**), indicating the homologous targeting capability of CM. Meanwhile, single CM modified UCNPs showed the limited capability to target and accumulate in the mitochondria, evidenced by abundant UCNPs in the cytoplasm rather than the mitochondria. Remarkably, the majority of HM-UCNPs were located in the mitochondria, suggesting the dual-targeting of HM to both tumour cells and mitochondria organelles.

Figure S6. TEM images of mitochondria in U87MG cells treated with HM-UCNPs, CM-UCNPs, MM-UCNPs and bare UCNPs. UCNPs: 1 mg/mL. Scale bar = 500 nm. The red arrows indicated the UCNPs.

3. *Figure 2i: what are the red arrows indicating?*

Response: We thank the reviewer for pointing this out, the red arrows indicated the damaged construction of mitochondria after treatment of nanomedicines. We have added this sentence in **Page 26 Lines 14:** The red arrows indicated the damaged mitochondria structure after treatment of nanomedicines.

4. *In vitro BBB model is oversimplified: astrocytes should be at least co-cultured with endothelial cells.*

Response: Thanks for the reviewer's constructive comment. As suggested, we have carried out the BBB penetration by seeding the endothelial cells and astrocytes in the upper chamber and glioblastoma in the lower chamber of transwell (J Control Release, 2018, 273:108-130). The results showed that both the HM-NPs@Cy5 and CM-NPs@Cy5 displayed the homologous targeting and BBB penetration ability with a higher penetration ratio of 7.5% than that of MM-NPs@Cy5 (4.8%) and NPs@Cy5 (3.7%) (**Figure 4d**). Similarly, the results of flow cytometry agreed with the transport ratios (**Figure 4f**). Importantly, the superiority of HM-NPs disappeared when treated with the inhibitors for maintaining the BBB integrity (**Figure S17**). These results have been included in **Page 11 Lines 18-26:** We determined if GBM cell membrane decorated NPs can traverse the BBB by modulating the tight junctions between endothelial cells using an *in vitro* BBB model consisting of a top layer of human cerebral

microvascular endothelial cells (hCMEC/D3) as well as astrocytes (HA1800 cells) and lower layer of U87MG cells. To assess the BBB model integrity, trans-endothelial electrical resistance (TEER) was continuously monitored after nanoparticles were added to the upper compartment. In this model, HM-NPs enhanced BBB traversal was observed (Figure 4d, 4e) with evidences that the TEER of the hCMEC/D3 and HA1800 bilayer decreased after HM-NPs or CM-NPs treatment (Figure 4f), whereas treatment with MM-NPs, naked NPs or PBS showed little, or no, reductions in TEER values.

Figure 4d, Illustration of the *in vitro* BBB model. **e**, Transport ratios of bare NPs, CM-NPs, MM-NPs or HM-NPs in hCMEC/D3 monolayer of *in vitro* BBB model (Cy5 concentration: 10 $\mu\text{g}/\text{mL}$). **f**, The trans-endothelial electrical resistance (TEER, $\Omega \cdot \text{cm}^2$) values in the *in vitro* BBB model at different time points after incubation with HM-NPs, MM-NPs, CM-NPs or NPs. **g**, The TEER ($\Omega \cdot \text{cm}^2$) values in the *in vitro* BBB model pretreated with cyclic adenosine monophosphate (cAMP) inhibitors (8-CPT-cAMP and Ro 20-1724) after incubation with HM-NPs, MM-NPs, CM-NPs or NPs.

Figure S17. BBB penetration and cellular uptake analysis in vitro. The fluorescence intensities were detected with flow cytometry of U87MG cells after nanoparticles traversed the BBB without (a) or with (b) pretreated with 8-CPT-cAMP and Ro 20-1724 (FITC concentration: 10 $\mu\text{g}/\text{mL}$).

5. ATP levels in cells should be evaluated upon the different treatments.

Response: As suggested, we have evaluated the ATP levels in U87MG and X01 cells upon the different formulations. The results showed that HM-NPs@G resulted in a sharp reduction of ATP levels in both U87MG and X01 cells (Figure 4f, S11). Correspondingly, we have added on the results in Page 8 Lines 2-3: HM-NPs@G resulted in a sharp reduction of ATP levels in both U87MG and X01 cells (Figure 2f, S11).

Figure 2f. ATP concentrations of U87MG cells when treated with HM-NPs@G, MM-NPs@G, CM-NPs@G, free Gboxin (Gboxin: 800 nM) and PBS for 72 h (n=3). Data are presented as mean \pm SD (one-way ANOVA and Tukey multiple comparisons tests).

Figure S11. ATP level of X01 cells when treated with HM-NPs@G, MM-NPs@G, CM-NPs@G, free Gboxin (Gboxin: 800 nM) and PBS for 72 h (n=3). Data are presented as mean \pm SD (one-way ANOVA and Tukey multiple comparisons tests).

6. Characterization (in terms of stemness markers) of glioma stem cells is missing.

Response: We thank the reviewer's nice comments. The glioblastoma stem cells (GSCs) are kindly provided by Prof. Jong Bae Park (National Cancer Center of South Korea) and the

stemness properties of the GSCs were verified by the previous work. To clarify, we have provided more details in **Page S3 Lines 24-25 & Page S4 Line 1**: The X01 glioblastoma stem cells (GSCs) were kindly provided by Professor Jong Bae Park from the National Cancer Center of South Korea, the more details are in the previous works which have been cited (Science Advances, 2022, 8, eabm8011; Cancer Research, 2017, 77, 18; PLOS Biology, 2015, 13, e1002152; Oncotarget, 2014, 5, 6756-6769).

REVIEWER COMMENTS

Reviewer #1 (Remarks to the Author):

I felt that the authors have adequately responded to my questions. However, the U87 model should be moved to the supplement and its limitations should be acknowledged.

Reviewer #2 (Remarks to the Author):

My comments have been addressed in this revised version.

Reviewer #3 (Remarks to the Author):

The authors have conducted additional experiments and added several statements as per the reviewer's comments. However, further improvements are necessary to solidify their responses. Detailed comments are provided below.

1. In response to comment #6, the authors stated that the drug loading efficiency was terribly low and even undetectable due to the unique chemical structure of Gboxin, which prevents direct loading into the hydrophobic part of the hybrid cell membrane. Nevertheless, in Figures 1h, S15, and 3d, the authors detected the content of Gboxin using high-performance liquid chromatography (HPLC), which is inconsistent with the previous description. Additionally, the necessity of using polymer nanoparticles to load Gboxin should be discussed. Furthermore, more experimental details about HPLC should be provided.
2. As per the author's description, polymer nanoparticles are biocompatible, biodegradable, and eliminated from the body. Moreover, the authors cited a study (J Control Release, 2021, 10, 334: 127-137) to support their claim that larger nanoparticles with sizes of 20-100 nm access the kidney by dissociating into smaller particles and passing through the glomerular filtration barrier. Accordingly, experiments are necessary to confirm this point.
3. Figure 3b showed that the fluorescence intensity in the tumor was significantly higher than in other tissues in the HM-NPs group, while the fluorescence intensity in the tumor ex vivo was remarkably lower than in the kidney. The authors should provide an explanation

for this phenomenon.

4. The authors should adjust the brightness of the green fluorescent channel in Figure 5h.

Reviewer #4 (Remarks to the Author):

Authors replied to most of the comments; however, in my opinion, it is not yet totally clear the mechanism at the base of the subcellular targeting. Are mitochondrial fusion proteins (mitofusin, atlastin, etc.) responsible of this targeting? Would it be possible to provide an experimental hint in this regard?

Point-by-Point Responses to Reviewers' Comments

We would like to thank the reviewers for their positive and constructive comments. We have conducted corresponding experiments and carefully revised our manuscript according to the reviewers' comments. Point-by-point responses are as below.

Reviewer #1 (Remarks to the Author):

I felt that the authors have adequately responded to my questions. However, the U87 model should be moved to the supplement and its limitations should be acknowledged.

Response: We appreciate the reviewer's constructive comment. As suggested, we have removed the results of U87MG model (original Figure 5) to the supplemental materials (Figure S20) and further discussed the limitations of U87MG model in the main text in **Page 14 Lines 1-6**: Although HM-NPs@G nanomedicines have shown good therapeutic effects on U87MG orthotopic mice models, it is admitted that U87MG model has a couple of limitations including the unclear originals. Therefore, the patient-derived xenograft (PDX) glioblastoma stem cells (GSCs) models are adopted as they are more closely resemble clinical GBM characteristics than standard GBM cell derived xenograft (CDX) models. Next, we investigated the anti-tumour effect of HM-NPs@G in X01 patient-derived GSCs xenograft models.

Reviewer #2 (Remarks to the Author): with expertise in glioblastoma, metabolism

My comments have been addressed in this revised version.

Response: We thank for the reviewer's positive comment.

Reviewer #3 (Remarks to the Author): with expertise in nanoparticles, glioblastoma

The authors have conducted additional experiments and added several statements as per the reviewer's comments. However, further improvements are necessary to solidify their responses. Detailed comments are provided below.

- 1. In response to comment #6, the authors stated that the drug loading efficiency was terribly low and even undetectable due to the unique chemical structure of Gboxin, which prevents direct loading into the hydrophobic part of the hybrid cell membrane. Nevertheless, in Figures 1h, S15, and 3d, the authors detected the content of Gboxin using high-performance liquid chromatography (HPLC), which is inconsistent with the previous description. Additionally, the necessity of using polymer nanoparticles to load Gboxin should be discussed. Furthermore, more experimental details about HPLC should be provided.*

Response: We sincerely apologize for the confusion. Actually, when we tried to load Gboxin with hybrid membrane without any other polymeric materials directly at the beginning, the drug loading efficiency was terribly low or even undetectable. It may be due to the fact that Gboxin could not be encapsulated by the phospholipid bilayer of membranes only with limited hydrophobic space. Therefore, it is necessary to develop polymeric nanocarriers to efficiently encapsulate and deliver Gboxin. Then, we tried various diblock polymers and screened out PEG-PHB as the optimal nanocarriers to deliver Gboxin for further studies as it possesses the highest drug loading content (DLC). We would like to explain that the HPLC results in Figure 1 h, S15 and 3 d showed that Gboxin could be loaded and responsively released by PEG-PHB polymeric nanocarriers, which does not conflict with our previous statements that single membrane could not load Gboxin efficiently.

As suggested, we have discussed the necessity of polymeric nanocarriers in loading Gboxin in the manuscript in **Page 16 Lines 22-29**: It should be noted that PEG-PHB polymeric nanocarriers used for delivering Gboxin into the brain are inevitable and necessary due to the following points: (1) the polymeric nanocarriers could effectively encapsulate the Gboxin to protect it from degradation during the blood circulation, significantly improving the extremely short circulation time of Gboxin. (2) the ROS-responsive PEG-PHB triggers the release of loaded Gboxin in tumour tissues and cells possess high level of ROS, while preventing the drug release in normal physiological conditions, leading to potent anticancer effects with few side effects. Therefore, it is indispensable to employ polymeric nanocarriers to deliver Gboxin in the GBM therapy.

Additionally, the concentration of Gboxin was detected by High Performance Liquid Chromatography (HPLC, Agilent G7129C). The analysis was performed on a Waters system with A: ddH₂O (0.01% TFA) and B: acetonitrile (0.01% TFA) as the eluent (5% to 95% B within 1.3 min, 95% to 5% B from 1.3 min to 3.0 min), flow rate: 1.8 mL min⁻¹, UV wavelength: 214/254 nm, injection volume: 10 µL, Column: SunFire C18 (50 mm × 4.6 mm, 3.5 µm), retention time: 1.4 min.

All the information is included in the supporting information in **Page S3 Lines 13-18**: The concentration of Gboxin was detected by High Performance Liquid Chromatography (HPLC, Agilent G7129C). The analysis was performed on a Waters system with A: ddH₂O (0.01% TFA) and B: acetonitrile (0.01% TFA) as the eluent (5% to 95% B within 1.3 min, 95% to 5% B from

1.3 min to 3.0 min), flow rate: 1.8 mL min⁻¹, UV wavelength: 214/254 nm, injection volume: 10 µL, Column: SunFire C18 (50 mm × 4.6 mm, 3.5 µm), retention time: 1.4 min.

2. *As per the author's description, polymer nanoparticles are biocompatible, biodegradable, and eliminated from the body. Moreover, the authors cited a study (J Control Release, 2021, 10, 334: 127-137) to support their claim that larger nanoparticles with sizes of 20-100 nm access the kidney by dissociating into smaller particles and passing through the glomerular filtration barrier. Accordingly, experiments are necessary to confirm this point.*

Response: We thank the reviewer's constructive comment. To investigate whether HM-NPs were metabolized and excreted out from the body by the renal glomerulus, we assessed the renal distribution of HM-NPs at three consecutive time points (1 h, 6 h, and 48 h) post injections. Interestingly, the results showed that HM-NPs localized inside renal corpuscles and became stronger from 1 h to 6 h, while weaker fluorescence was observed in the renal glomerulus at 48 h. Take into account that HM-NPs firstly accumulated in the renal corpuscles, and then eventually faded away over time. Accordingly, HM-NPs could be metabolized and excreted out through the glomerulus of the kidney.

Accordingly, relative descriptions were added in the main text in **Page 11 Lines 1-3**: **Given the high levels of HM-NPs detected in the kidney, we considered that HM-NPs could be metabolized and excreted out of the body by renal corpuscles, which further, at least partly confirmed by the results of renal distribution of HM-NPs (Figure S18).**

Figure S18. Tissue-level distribution in renal corpuscles of HM-NPs. Immunofluorescence images of kidney taken from healthy Balb/C mice at 1 h, 6 h, and 48 h post injection of HM-NPs@Cy5 (2 mg Cy5 equiv. kg⁻¹). (Nuclei were stained with DAPI (blue) and renal glomerulus with Anti-Nephrin Rabbit pAb (green); Cy5 fluorescence is red. Scale bars = 100 μm.

3. *Figure 3b showed that the fluorescence intensity in the tumor was significantly higher than in other tissues in the HM-NPs group, while the fluorescence intensity in the tumor ex vivo was remarkably lower than in the kidney. The authors should provide an explanation for this phenomenon.*

Response: We thank the reviewer for pointing this out. It is true that HM-NPs have stronger Cy5 intensity in brain tumors than the other tissues in *in vivo* imaging (Figure 3b), it may be attributed to the fact that HM-NPs located in blood vessels and tumour tissues which were both captured by the IVIS imaging machine. However, the tissues in *ex vivo* imaging were conducted after washing and perfusion to eliminate the HM-NPs in the blood, causing weaker Cy5 fluorescence in tumours compared with that of the *in vivo* imaging results.

4. *The authors should adjust the brightness of the green fluorescent channel in Figure 5h.*

Response: We appreciate the reviewer's nice comment, we have adjusted and normalized the green fluorescence to make the images more clear in Figure S20h (original Figure 5h).

Figure S20h. Histological analysis using the TUNEL assay. Green: apoptotic cells; blue: Hoechst-stained cell nuclei (Scale bar = 100 μ m).

Reviewer #4 (Remarks to the Author): with expertise in nanoparticles, glioblastoma

Authors replied to most of the comments; however, in my opinion, it is not yet totally clear the mechanism at the base of the subcellular targeting. Are mitochondrial fusion proteins (mitofusin, atlastin, etc.) responsible of this targeting? Would it be possible to provide an experimental hint in this regard?

Response: We appreciate the reviewer's constructive comment. To reveal the subcellular targeting mechanism of HM-NPs@G, we selected a mitofusin inhibitor, MFI8, to evaluate the Gboxin content in mitochondria. The results showed that the accumulation of HM-NPs@G in mitochondria isolated from U87MG cells pre-treated with MFI8 remarkably reduced compared to the group without pre-treatment, suggesting that mitofusin plays a key role in targeting and penetrating mitochondria of HM-NPs (Figure S7). Accordingly, we have added the result in **Page 7 Lines 7-11:** Moreover, to reveal the subcellular targeting mechanism of HM-NPs@G, we selected a mitofusin inhibitor MFI8 to evaluate the Gboxin content in mitochondria. The results showed that the accumulation of HM-NPs@G in mitochondria isolated from U87MG cells pre-treated with MFI8 remarkably reduced compared to the group without pre-treatment, suggesting that mitofusin plays a key role in targeting and penetrating mitochondria of HM-NPs (Figure S7).

Figure S7. Mitochondria targeting and penetrating analysis of HM-NPs@G *in vitro*. The content of Gboxin in mitochondria isolated from U87MG cells, with or without MFI8 pre-treatment, followed by incubated with HM-NPs@G for 24 h (Gboxin: 20 $\mu\text{g mL}^{-1}$, $n = 3$). Free MFI8 was pre-incubated with U87MG cells for 6 h (MFI8: 5 μM). The Gboxin accumulation of HM-NPs@G with pre-treatment is relative to the HM-NPs@G without pre-treatment. Data are presented as mean \pm SD (Statistical analysis was performed using two-sample t-test).

REVIEWERS' COMMENTS

Reviewer #3 (Remarks to the Author):

My comments have been well addressed in this revised version.

Reviewer #4 (Remarks to the Author):

Most of the comments have been addressed, despite the mechanisms would deserve, in my opinion, a better characterization.

A general language/style revision is recommended.

Point-by-Point Responses to Reviewers' Comments

We would like to thank the reviewers for their time and constructive comments on our manuscript. According to the comments, we have carefully improved our manuscript. Below the comments of reviewers are responded point-by-point.

Reviewer #3 (Remarks to the Author):

My comments have been well addressed in this revised version.

Response: We thank the reviewer for the positive comment.

Reviewer #4 (Remarks to the Author):

Most of the comments have been addressed, despite the mechanisms would deserve, in my opinion, a better characterization.

A general language/style revision is recommended.

Response: We appreciate the reviewer's constructive comments. The mitochondria targeting mechanism is very complicated and multiple proteins and pathways are involved. Taking into account of the current methods, we currently characterize the mitochondria targeting mechanism by inhibiting mitofusin protein as it plays a key role in the fusion of mitochondria membrane (*Trends Cell Biol*, 2021, 31: 62-74; *Nat Commun*, 2022, 13: 3775), while other mitochondria membrane proteins that may be related to the target capability have not been found. Our results show that the accumulation of HM-NPs@G in mitochondria is decreased significantly after inhibiting the mitofusin protein, suggesting that mitofusin plays an important role in targeting and penetrating mitochondria. In our future work, we will continually investigate the mitochondria targeting mechanism. Accordingly, we have further discussed the mitochondria targeting mechanisms in **Page 7 Lines 10-18: Moreover, to reveal the subcellular targeting mechanism of HM-NPs@G, we selected a mitofusin inhibitor MFI8 to evaluate the Gboxin content in mitochondria as the mitofusin protein has been reported to play a key role in fusion of mitochondria membrane and mainly expresses on the MM. The results show that the accumulation of HM-NPs@G in mitochondria isolated from U87MG cells pre-treated with MFI8 remarkably reduced compared to the group without pre-treatment, suggesting that mitofusin plays a key role in targeting and**

penetrating the mitochondria (**Supplementary Figure 7**). However, the mitochondria-targeting mechanism is very complicated. We briefly demonstrate that mitofusin is involved in the mitochondria targeting of HM-NPs, and the systematic targeting mechanism deserves further investigation.

Furthermore, we have polished the language and style of our main text as required.